# Principal cell activity induces spine relocation of adult-born interneurons in the olfactory bulb

Vincent Breton-Provencher[1], Karen Bakhshetyan[1,*], Delphine Hardy[1,*], Rodrigo Roberto Bammann[1], Francesco Cavarretta[2,3], Marina Snapyan[1], Daniel Côté[1,4], Michele Migliore[2,3,5] & Armen Saghatelyan[1,6]

Adult-born neurons adjust olfactory bulb (OB) network functioning in response to changing environmental conditions by the formation, retraction and/or stabilization of new synaptic contacts. While some changes in the odour environment are rapid, the synaptogenesis of adult-born neurons occurs over a longer time scale. It remains unknown how the bulbar network functions when rapid and persistent changes in environmental conditions occur but when new synapses have not been formed. Here we reveal a new form of structural remodelling where mature spines of adult-born but not early-born neurons relocate in an activity-dependent manner. Principal cell activity induces directional growth of spine head filopodia (SHF) followed by spine relocation. Principal cell-derived glutamate and BDNF regulate SHF motility and directional spine relocation, respectively; and spines with SHF are selectively preserved following sensory deprivation. Our three-dimensional model suggests that spine relocation allows fast reorganization of OB network with functional consequences for odour information processing.

[1] Cellular Neurobiology Unit, Institut Universitaire en santé mentale de Québec, Quebec City, Quebec, Canada G1J 2G3. [2] Department of Mathematics, University of Milan, Milan 20133, Italy. [3] Department of Neurobiology, Yale University School of Medicine, New Haven, Connecticut 06520, USA. [4] Centre d'optique, photonique et laser (COPL), Université Laval, Quebec City, Quebec, Canada G1V 0A6. [5] Institute of Biophysics, National Research Council, Palermo 90146, Italy. [6] Department of Psychiatry and Neuroscience, Université Laval, Quebec City, Quebec, Canada G1V 0A6. * These authors contributed equally to this work. Correspondence and requests for materials should be addressed to A.S. (email: armen.saghatelyan@fmed.ulaval.ca).

The olfactory bulb (OB) is one of the few regions in the adult brain that displays a high level of structural plasticity due to a constant supply of adult-born periglomerular and granule cells (GC)[1,2]. GC form dendro-dendritic reciprocal synapses with the lateral dendrites of principal cells (mitral cells (MC)). The inhibition provided by these synapses synchronizes the activity of MC, allowing for fine spatio-temporal tuning of their responses to odours[3]. Adult-born GC play an important role in this process[4], and several studies have shown that they are involved in short-[4] and long-term odour memory[5], odour discrimination[6] and social behaviour[7]. The central role played by adult-born neurons in different odour-dependent tasks is likely due to their increased responsiveness to odour stimulation[8], as well as to transient experience-dependent synaptic modifications at proximal glutamatergic fibre–GC synapses[9].

The continuous supply of new neurons provides the OB with a reservoir of plastic cells that enables synaptic remodelling due to the continuous formation, elimination and/or stabilization of new spines of immature[10–14] and mature adult-born neurons[15–17]. Adult-born GCs thus constantly sculpt the bulbar network in response to changing environmental conditions. However, environmental changes can be rapid, whereas the synaptogenesis of adult-born neurons occurs over a longer time scale. It is thus conceivable that the OB network requires considerably quicker structural modifications to existing dendro-dendritic synapses to adapt the functioning of the OB network to changing environmental conditions.

In the present study, we used *in vivo* and *in vitro* two-photon time-lapse imaging, with a relatively rapid acquisition rate (once every 5 min), to show that synaptic remodelling in the OB network also relies on the relocation of the spines of mature adult-born neurons. Chronic *in vivo* imaging revealed that relocated spines are stabilized in the bulbar network and are directly opposed to synaptophysin-labelled puncta. We linked spine relocation to the growth of thin filopodia-like protrusions from the spine head (spine head filopodia (SHF)) that depended on the level of odour-induced activity. Stimulation of a single MC induced directional growth of SHF towards activated MC dendrites that in turn promoted spine relocation. Glutamate released from MC induced the AMPA receptor (AMPAR)-dependent motility of SHF, whereas the activity-dependent release of BDNF by MC drove directional growth of SHF and spine displacement. On the other hand, there were fewer SHF on the spines of early-born GC, and they were unaffected by odour stimulation. Our findings suggested that the spines of adult-born neurons can relocate from inactive to active principal cell dendrites, highlighting a new form of structural plasticity in the constantly remodelling OB network. Furthermore, our computational modelling experiments suggested that the relocation of mature spines allows rapid adjustment of the OB network to odour-induced activity that can have functional consequences for odour information processing.

## Results

**Dendritic spines of adult-born GC relocate in the OB network**. To study the structural remodelling of adult-born neurons over relatively short time intervals, we performed *in vivo* two-photon time-lapse imaging in the adult OB. Neuronal precursors were labelled by the stereotaxic injection of a GFP-encoding lentivirus into the adult rostral migratory stream (RMS) (Fig. 1a) and time-lapse imaging of adult-born GC in the OB was performed every 5 min for 60–240 min (Fig. 1b,c). Our experiments revealed that adult-born GC undergo a novel form of structural remodelling via the relocation of some spines on the apical part of the dendrites (Fig. 1d). In some cases, the spines relocated by at least 2–4 μm (Fig. 1d,h; Supplementary Movie 1). Spine relocation was rapid and occurred within a few minutes (Fig. 1d,h; Supplementary Movie 1; Supplementary Fig. 1). The relocation of the spines was observed at 30–50 and 120–150 days post-injection (d.p.i.) of the viral vector into the RMS, indicating that the relocating spines were located on fully integrated and mature adult-born GC.

In addition, spine relocation was observed in acute OB sections at 14, 28, 42 and >77 d.p.i. (163 d.p.i. being the longest time point studied) (Fig. 1e–h; Supplementary Fig. 1a; Supplementary Movie 2), confirming our results in more stable imaging conditions than *in vivo*. To quantify the percentage of relocating spines and the extent of their relocation, we tracked 514 randomly selected spines at different d.p.i. and identified relocating spines based on the Z-score of their direction vector amplitude, which is a measure of the directionality of spine relocation (Supplementary Fig. 1b,c). Under baseline (BL) conditions, 4.5% of the spines relocated above the threshold (Fig. 1i). This percentage was similar at different d.p.i. (range 3–8% for 14, 28, 42 and 77 d.p.i. *in vitro* and 30–60 d.p.i. *in vivo*). The mean ratios of spine relocation and of the amplitudes of the direction vectors were $1.2 \pm 0.2$ and $1.26 \pm 0.09$ μm, respectively ($n = 21$ spines from 18 cells, 16 mice) for spines that significantly relocated (above Z-score). The direction vector amplitudes for relocating spines were 10-fold higher than those for dendrites measured in the same images ($1.26 \pm 0.09$ μm for relocating spines versus $0.12 \pm 0.006$ μm for dendrites; Supplementary Fig. 1c,d), indicating stable imaging conditions and that spine relocation values are not influenced by measurement noise. Spine relocation mostly occurred as a lengthening of the spine neck, which is often associated with changes in orientation. Specifically, 16 of 21 relocating spines showed clear neck elongation, of which nine displayed elongation and changes in orientation. Three spines displayed changes in orientation only, and two displayed neck retraction.

Several key features distinguish spine relocation from other forms of structural modifications such as spine neck or spine volume plasticity. First, spine neck/volume changes observed after long-term potentiation (LTP) occur in the nanometre range[18,19], whereas the spine relocation we observed occur in the order of several micrometres and was as large as 2–4 μm (Fig. 1c–f). Second, spine neck plasticity has been described as a shortening and widening of the spine neck[18,19], whereas, in the present study, most of spine relocation resulted from a lengthening of the spine neck (16 of 21 spines). Third, spine relocation was often associated with changes in orientation (12 spines out of 21), which is not observed during spine neck plasticity induced by LTP[18,19]. Last, spine relocation was not associated with changes in spine volume (Supplementary Fig. 1e–g). Our data thus suggested that the structural plasticity in the OB network depends not only on the continuous arrival of new neurons but also on a new form of structural remodelling via the relocation of mature spines of adult-born GC.

**Spine relocation is preceded by SHF growth**. Our *in vivo* and *in vitro* two-photon time-lapse imaging revealed that spine relocation is often preceded by the growth of thin filopodia-like protrusions from the spine head, hereafter called SHF (Fig. 1d,f). The SHF were on average $2.1 \pm 0.1$ μm in length ($n = 97$ SHF, three cells, three mice). Based on 60 min of BL recordings, ~45% of the spines both *in vivo* and *in vitro* displayed SHF, several of which arose from the same spine and grew in different directions (Figs 1f and 2a,b). SHF were also observed on adult-born GC spines in fixed tissues (Fig. 2c). The percentage of spines with SHF was similar in fixed tissues and in snapshot images of

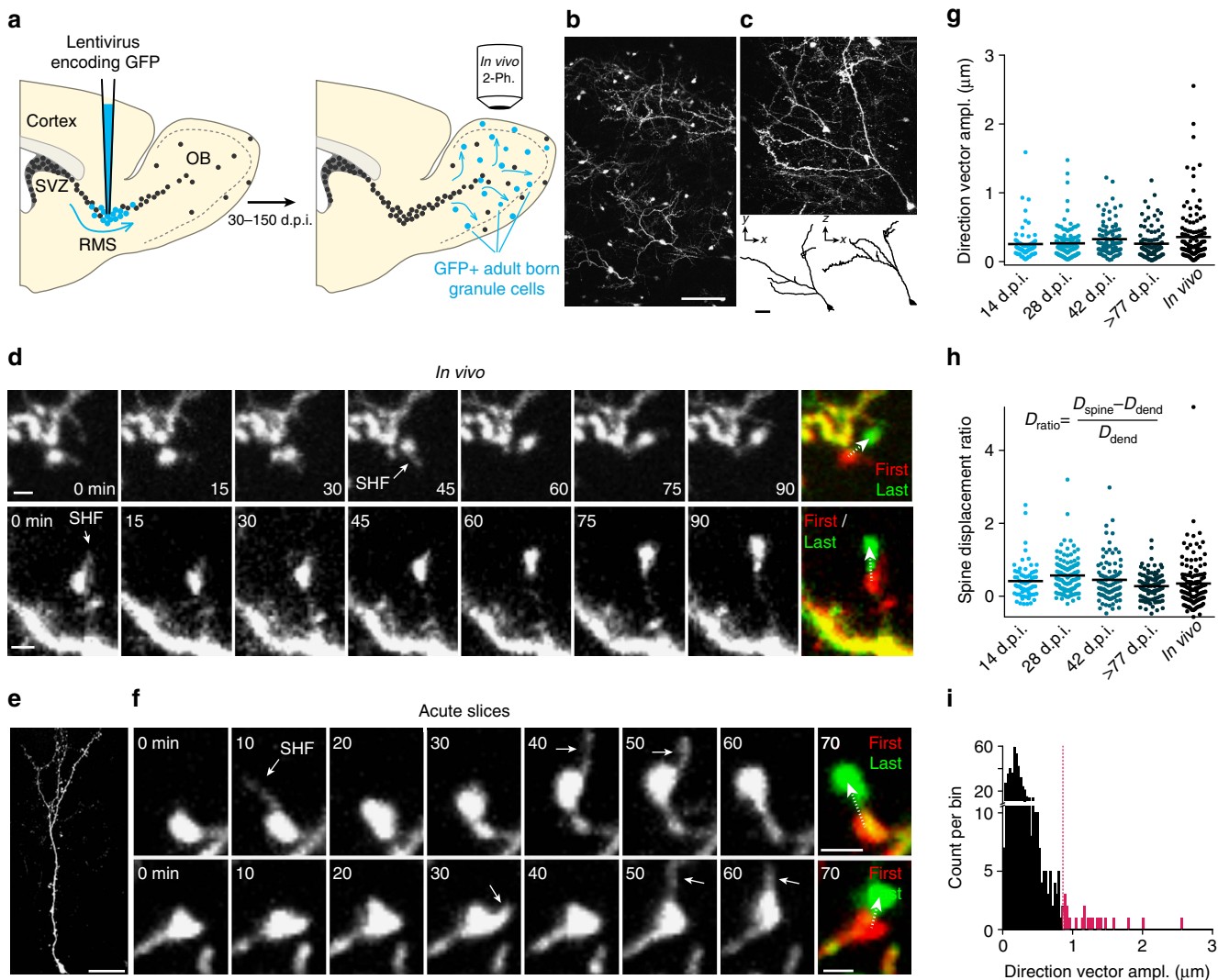

**Figure 1 | Relocation of mature spines of adult-born GC in the OB. (a)** Protocols used for labelling adult-born GC and for *in vivo* two-photon imaging. Neuronal precursor cells were infected with GFP-expressing lentiviral particles in the rostral migratory stream (RMS). Thirty to 150 days post-injection (d.p.i.), the spines of GFP + cells were imaged *in vivo* by two-photon microscopy through a cranial window. SVZ, subventricular zone. **(b)** Low-magnification projection image of a population of adult-born GC and periglomerular cells taken *in vivo*. **(c)** Top—projection image of an isolated adult-born GC. Bottom—lateral view (left) and top view (right) of the reconstructed neuron shown above. **(d)** Example from two adult-born GC dendrites imaged *in vivo* at 60–70 d.p.i. showing spine head relocation. The white arrows indicate spine head filopodia (SHF). The first and last z-stacks have been false-coloured in red and green to highlight the extent of spine relocation. **(e)** Example of an adult-born GC imaged in an acute slice. **(f)** Sequences of time-lapse two-photon images showing spine relocation on 14 d.p.i. (upper panels) and 42 d.p.i. (lower panels) adult-born GC distal dendrites. **(g,h)** Ratio of total spine displacement and direction vector amplitude that shows the directionality of spine relocation over 95 min for spines monitored at different maturational stages (*n* = 68, 124, 93, 103 and 126 spines from 7, 7, 7, 5 and 16 mice for the 14, 28, 42, 77 d.p.i. and *in vivo* conditions, respectively). **(i)** Histogram showing the direction vector amplitudes for 514 randomly selected spines. Spines with direction vector amplitudes above a z-score value are shown in red. Scale bars, 100 μm, 20 μm and 50 μm for **b,c,e**, respectively. The scale bar for **d,f** is 2 μm.

time-lapse movies (7.7 ± 0.9% of the spines with SHF in fixed tissues (*n* = 28 cells, 3 mice) and 7.6 ± 1.2% in acute OB slices (*n* = 16 cells, 10 mice). The total distance of spine relocation, as well as the directional vector were higher for the spines with SHF compared with the spines devoid of SHF (Fig. 2d,e), and 95.3% of the spines relocating above the threshold had SHF (20 of 21 spines, *n* = 16 mice; red dots on Fig. 2d). Of the 344 spines with at least 1 SHF (*n* = 57 cells, 42 mice), 6.1% relocated above threshold (range 4.5–13.5% for 14, 28, 42 and 77 d.p.i. *in vitro* and 30–60 d.p.i. *in vivo*), whereas only 1 of 160 spines devoid of SHF (0.6%) relocated above the threshold (*n* = 57 cells, 42 mice). To determine whether the direction of SHF growth determined the direction of spine relocation, we calculated the average vector

of SHF growth (Fig. 2f,g; black arrow) for relocating spines (red dots on Fig. 2d). Our analysis suggested that spines relocate towards the position where SHF had grown (Fig. 2g,h; compare red and black solid arrows). These results indicated that spine relocation is linked to the presence of SHF and that the direction of SHF growth determines the direction of spine relocation.

We then looked at whether spine relocation occurs on the GC dendritic segments showing increased structural dynamics. We examined the dynamics of SHF on ~100-μm-long dendritic segments centred around relocating (above a Z-score) and stable (without SHF) spines. Our analyses revealed higher structural dynamics on dendritic segments close to relocating spines (<15 μm) than those at remote locations (>15 μm) of the

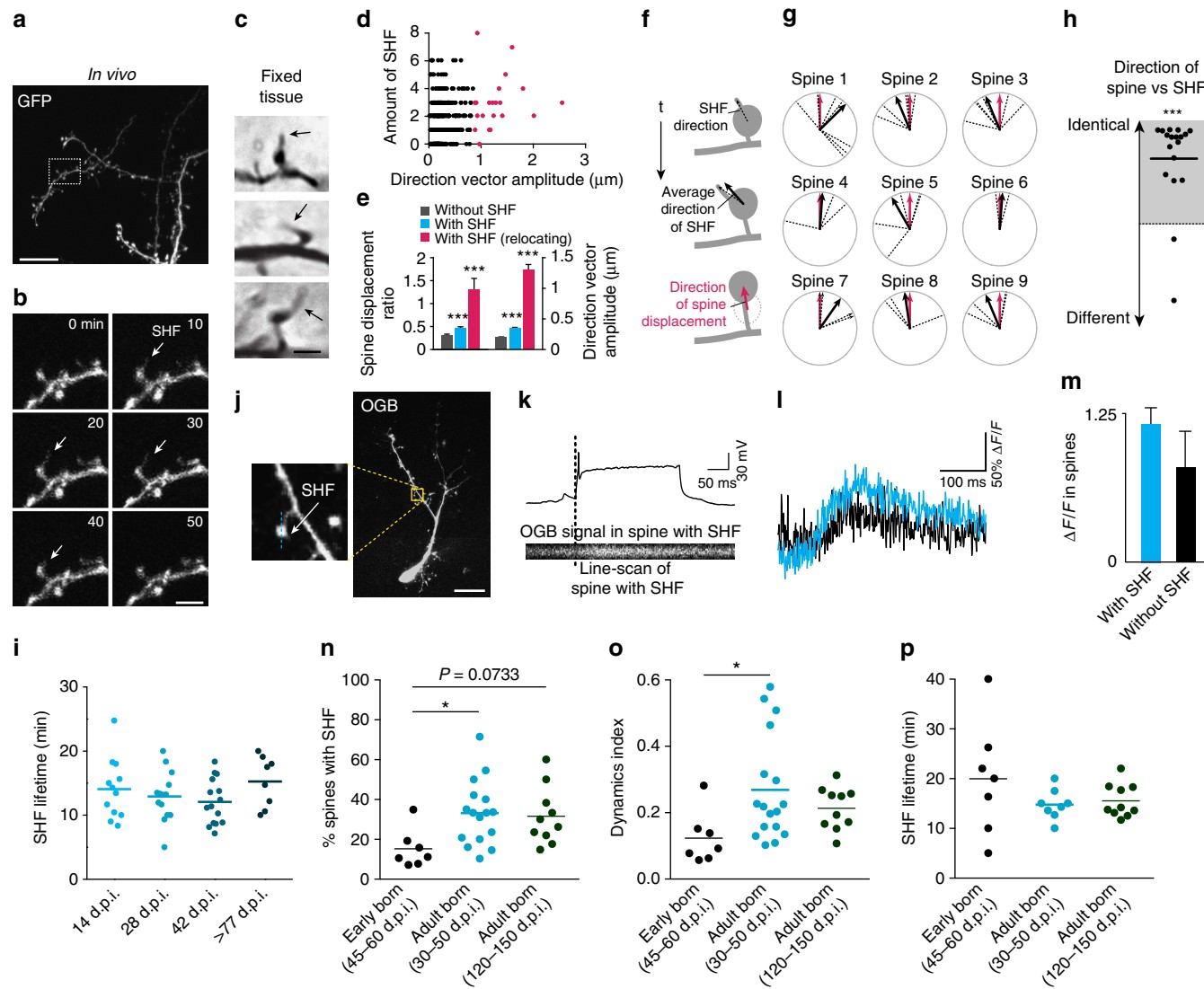

**Figure 2 | SHF determines the relocation of the spines of adult-born but not early-born GC.** (**a**) Low-magnification image of an adult-born GC used for *in vivo* two-photon imaging. (**b**) Image sequence of the boxed region in **a**. The arrow indicates SHF. (**c**) GFP + adult-born GC from fixed OB sections showing the presence of SHF. (**d**) Relationship between the number of SHF and the amplitude of the spine relocation direction vector ($n = 505$ spines). (**e**) Normalized spine displacement and amplitude of direction vector for spines with and without SHF ($n = 161$ and $344$ spines, respectively). (**f**,**g**) Measurement of the correlation between the direction of SHF protrusion and the direction of spine relocation. Each dashed black line represents individual SHF, whereas solid black and red arrows represent the average direction of SHF protrusion and the direction of spine relocation, respectively. (**h**) The difference between the average angle of SHF protrusion and the spine displacement vector was calculated for spines showing relocation above the Z-score ($n = 21$ spines, highlighted in red in **d**; ***$P = 2.88 \times 10^{-5}$ using a one-sample t-test). (**i**) SHF lifetime values for different maturational stages of adult-born GC ($n = 12$, 14, 15 and 8 cells from 10, 10, 9 and 5 mice for the 14, 28, 42 and 77 d.p.i. condition, respectively). (**j**) Example of an Oregon Green BAPTA (OGB)-filled td-tomato-expressing adult-born GC at 30 d.p.i. (**k**) Line scan $Ca^{2+}$ imaging of a GC spine with SHF (lower panel) following single action potentials induced by the depolarization of the GC (upper panel). (**l**,**m**) Individual traces and quantification of $Ca^{2+}$ transients in spines with (blue) and without (black) SHF ($n = 12$ and 13 spines with and without SHF, 8 cells, 8 mice). (**n**–**p**) Quantification of the percentage of spines with SHF and SHF dynamics and lifetimes for a period of 1 h of *in vivo* two-photon imaging for early-born at 45–60 d.p.i. and adult-born GC at 30–50 and 120–150 d.p.i. ($n = 7$, 16 and 10 cells from 6, 6 and 4 mice, respectively; *$P < 0.05$ using one-way ANOVA). Scale bars: 20 µm, 5 µm, 2 µm for **a–c**, respectively; 20 µm for **j**. ANOVA, analysis of variance.

same dendrite, as well as on dendrites with non-relocating spines (Supplementary Fig. 2d–f).

Since the relocation of spines on adult-born GC was maintained at all maturational stages studied, we then determined the dynamics and lifetimes of SHF at 14, 28, 42 and > 77 d.p.i. We observed no differences in SHF lifetimes and the number of appearing and retracting SHF (Fig. 2i and Supplementary Fig. 2a–c), suggesting that SHF on adult-born spines appear at 14 d.p.i. and remain beyond 77 d.p.i. To show that adult-born GC spines with

SHF are mature, we determined whether markers of mature spines, including PSD95 and synaptoporin, the latter being expressed by GC but not MC[14,20], are expressed by these spines. Our findings showed that these spines did indeed express PSD95 ($93.2 \pm 2.3\%$) and synaptoporin ($85.2 \pm 7.4\%$) ($n = 26$ cells, three mice). SHF were devoid of these markers, which is consistent with their rapid formation and retraction. To provide further evidence that spines with SHF are fully functional, we performed $Ca^{2+}$ imaging of spines with and without SHF following the induction

of back-propagating action potential. We first filled td-tomato virus-labelled adult-born GC with Oregon Green BAPTA (OGB), a $Ca^{2+}$ indicator, using a patch pipette (Fig. 2j). We then performed line scan imaging of spines with and without SHF following back-propagating action potentials induced by GC

depolarization via a patch pipette (Fig. 2k). Spines with SHF produced robust transient $Ca^{2+}$ signals. These transients were slightly higher, albeit not significantly, than those produced by spines without SHF, indicating that spines with SHF are fully functional (Fig. 2l–m).

We then determined whether spine relocation is specific to adult-born GC or whether it can also be observed in early-born interneurons. We thus labelled early-born GC by stereotaxic injection of a GFP-encoding lentivirus into the RMS of P10 pups and performed *in vivo* two-photon imaging in the OB 45–60 days later. No spine relocation of early-born GC was observed. Moreover, early-born GC had a lower percentage of spines with SHF, a lower SHF dynamics and a tendency for higher lifetime than their adult-born counterparts (Fig. 2n–p). Altogether, these results suggested that the SHF growth vector determines the position of spine relocation and that this new form of structural plasticity of mature, fully functional spines in the OB network is fulfilled by adult-born but not by early-born GC.

**Relocated spines are maintained in the OB network**. To determine whether relocating spines are maintained in the OB network, we performed chronic *in vivo* two-photon imaging. We first imaged spines of adult-born 30–60 d.p.i. GC to identify relocating spines and retrieved them 24–48 h later during a second imaging session (Fig. 3a,b). The mean direction vector amplitude of relocated spines during the first imaging session was $0.96 \pm 0.2\,\mu m$ ($n = 15$ spines from 11 mice). Interestingly, 93.3% of these spines were found on the second imaging day, indicating that relocating spines are stabilized and maintained in the OB network (Fig. 3c,d). In comparison, 90.7% of non-relocating spines were found on the second imaging day ($n = 43$ spines from 11 mice). Spine relocation was observed on the apical dendrites of adult-born GC in the external plexiform layer where they receive exclusive input from bulbar principal neurons[21]. We then investigated whether the relocated spines were part of dendro-dendritic synapses with MC dendrites. To address this issue, we identified relocated spines by *in vivo* two-photon imaging and then performed immunolabeling in the fixed tissue to detect synaptophysin, which is expressed by both MC and GC spines[20]. Our analysis revealed that relocating spines express synaptophysin and are directly opposed to synaptophysin-positive puncta (Fig. 3e). This result was consistent with the dendro-dendritic nature of GC apical dendrite synapses and suggested that relocated spines are part of synaptic contacts with MC.

**MC activity induces SHF growth and spine relocation**. Since several SHF may emerge from the same spine and grow in different

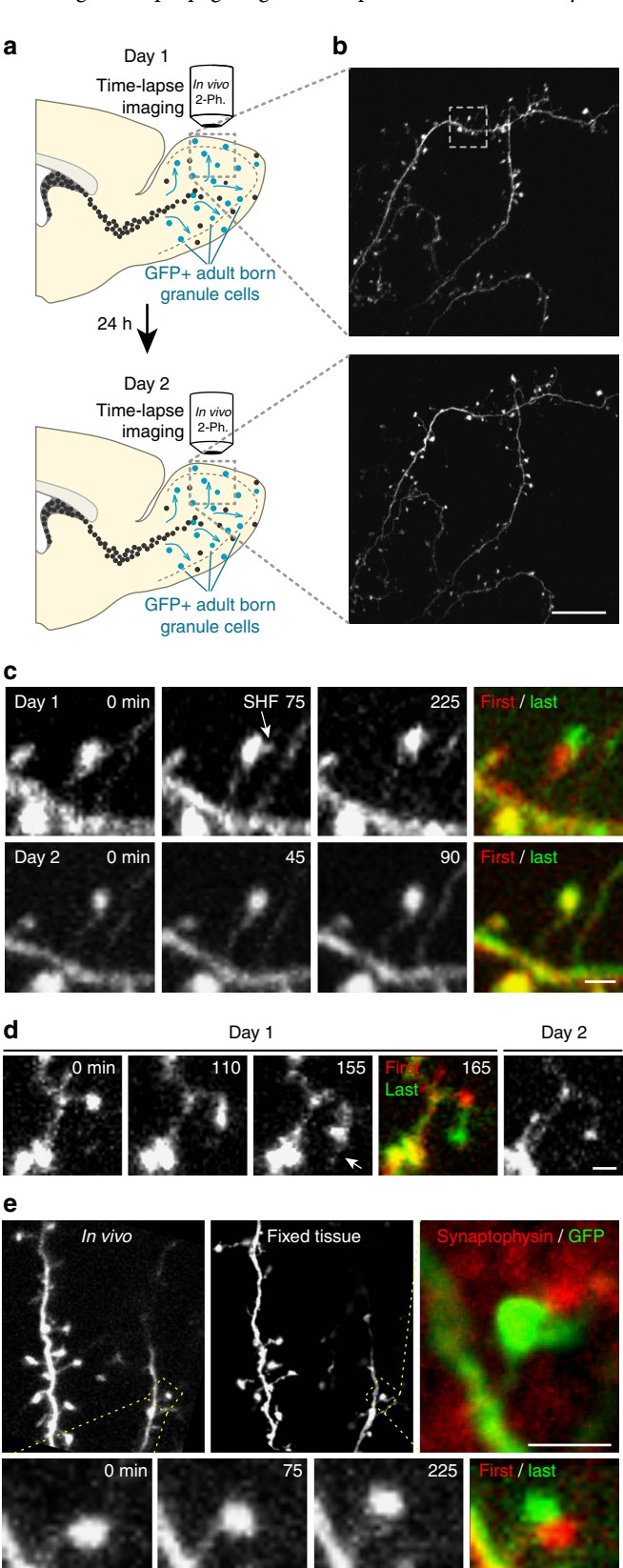

**Figure 3 | Relocated spines are stabilized in the OB network and are part of the dendro-dendritic synapses.** (**a**) Experimental procedure for chronic *in vivo* two-photon imaging. (**b**) Example of GC dendrite images acquired on 2 consecutive days. (**c**) Higher magnification of the boxed area in **b**. Time-lapse two-photon imaging of relocating spine during the first imaging day (upper panel). Note that the same spine is stabilized at the relocated position and that no displacement is observed during the second imaging day. The first and last z-stacks have been false-coloured in red and green to compare the first and last images of the time-lapse sequence, respectively. (**d**) Another example of relocating spine. (**e**) *In vivo* images showing the spine relocation of an adult-born GC (lower panels). At the end of the *in vivo* imaging, the animal was perfused, and OB sections were prepared for anti-synaptophysin immunohistochemistry. Confocal image showing the dendrite of the imaged GC (upper middle panel) and the anti-synaptophysin labelling (upper right panel). Note that spine relocating *in vivo* is directly opposed to synaptophysin puncta. Scale bars: 20 μm for **b**; 2 μm for **c,d**; 5 μm for **e** centre panel and 2 μm for **e** right panels.

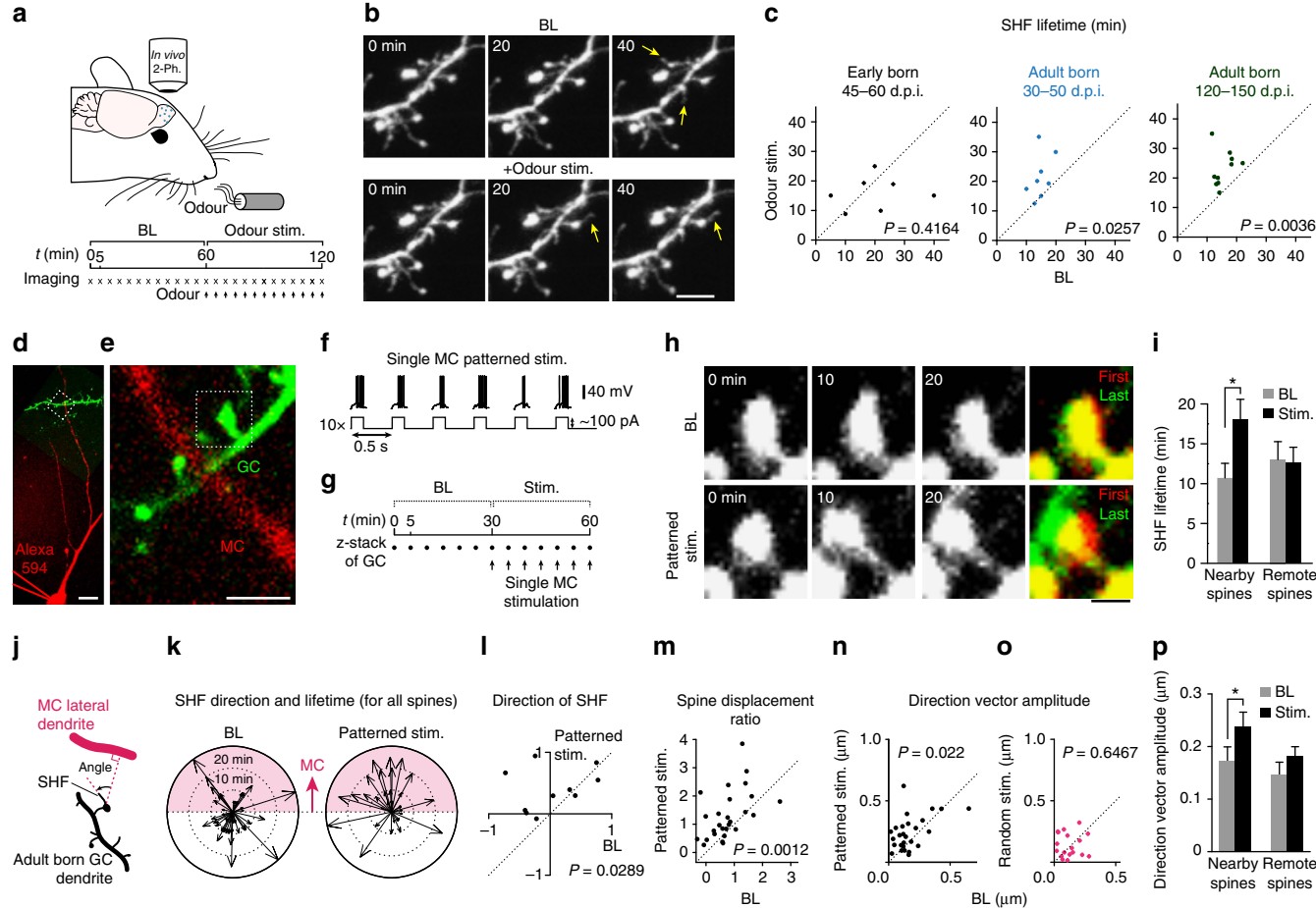

**Figure 4 | Olfactory sensory activity stabilizes SHF and promotes spine relocation. (a,b)** Experimental procedure and time-lapse imaging during sensory stimulation *in vivo*. Scale bar, 5 μm. **(c)** Quantification of effects of sensory stimulation on the SHF lifetime *in vivo* for early-born GC at 45–60 d.p.i. and adult-born GC at 30–50 and 120–150 d.p.i. ($n = 7$, 8 and 7 cells from 6, 6 and 4 mice, respectively). **(d)** Example of Alexa594-filled MC (red) and adult-born GC (green) dendrites. Scale bar, 20 μm. **(e)** Spines of an adult-born GC located within 5 μm of MC dendrite. Scale bar, 5 μm. **(f,g)** Stimulation pattern used to reproduce MC activity in response to an odour, and the experimental design. The top traces in **f** show an example of an MC recorded during stimulation. **(h)** Image sequence of the spine highlighted in **e** before (baseline) and during (Stim) the MC stimulation. Scale bar, 2 μm. **(i)** Comparison of the effect of MC stimulation on the SHF lifetimes of spines located within 5 μm and >10 μm away from the stimulated lateral dendrites. **(j)** Illustration of the method used to calculate the SHF direction based on the position of the MC lateral dendrite. **(k)** Graph showing the direction of the growth and lifetimes of the SHF. The arrows indicate the orientation of the SHF with respect to the dendrite of the MC. The lengths of the arrows represent the SHF lifetimes ($n = 43$ and 32 SHF for baseline and stimulation conditions, respectively). **(l)** Quantification of the cosine of the angle between the SHF and the MC lateral dendrite for each spine analysed before and during MC stimulation. **(i–l)**: $n = 11$ MC–GC pairs, 9 mice. **(m,n)** Quantification of the normalized spine displacement and direction of amplitude vector at baseline and during MC stimulation ($n = 27$ spines, 11 MC–GC pairs, 9 mice). **(o)** Random stimulation pattern delivered to a single MC did not induce spine relocation ($n = 18$ spines from nine MC–GC pairs, nine mice). **(p)** Comparison of the effect of MC stimulation on spine relocation for spines close to or far from MC dendrite. **(m,n)** $*P < 0.05$ with paired Student's *t*-test.

directions under BL conditions, we hypothesized that SHF on adult-born GC spines may be important in sensing the OB micro-environment and that their dynamics might depend on odour stimulation. To test this hypothesis, we performed *in vivo* two-photon imaging of adult-born and early-born GC during odour stimulation (Fig. 4a,b). We presented, via an olfactometer, a mixture of butyraldehyde and methylbenzoate, which is known to activate the medio-lateral region of the dorsal surface of the OB[22]. Our experiments revealed that odour stimulation stabilizes the SHF of adult-born GC at both 30–50 and 120–150 d.p.i. (Fig. 4c). Interestingly, however, the odour stimulation had no effect on the SHF dynamics of early-born GC (Fig. 4c), which again implies that adult-born GCs make a specific contribution to the structural remodelling of the bulbar network over a relatively rapid time frame.

We then investigated whether SHF are preferentially guided towards activated MC dendrites that in turn induce spine relocation. To directly address this question, we selectively stimulated single MC in acute OB slices, avoiding the effect of broad activation of the bulbar network induced by odour stimulation. We filled the MC with Alexa594, and imaged SHF dynamics on GFP-labelled adult-born GC spines positioned close to (5 μm) and far away (>10 μm) from the Alexa594-filled MC dendrite (Fig. 4d,e). The MC was stimulated with two different patterns, one that mimicked the activity of these cells induced by odour presentation (hereafter called physiological pattern stimulation; Fig. 4f,g)[23] and one that consisted of the same number of spikes given in a random order (hereafter called random stimulation, Supplementary Fig. 3). The images under control conditions were taken over a 30-min period, followed by 30 min of imaging during the stimulation of the MC (Fig. 4g). Consistent with our *in vivo* results following odour stimulation, the SHF lifetimes of spines located close to the stimulated MC

dendrite were significantly longer than those located far away after physiological pattern MC stimulation (Stim) compared with the BL condition (Fig. 4h,i). Interestingly, the tracking of each SHF with respect to the MC dendrite location and the calculation of the cosines of the angle between them showed that SHF growth became highly directional towards the stimulated MC dendrite (Fig. 4j-l). We then determined whether the directional growth of SHF towards stimulated MC dendrites is followed by spine relocation. Our analysis revealed a significant increase in the ratio of spine displacement (Fig. 4m), and the amplitude of the direction vector (Fig. 4n,p) after physiological MC stimulation. Moreover, stimulating the MC with a physiological pattern increased the percentage of spines with SHF that relocated above the threshold from 3.7 to 25.9% (30 min imaging under BL and stim. conditions, $n = 27$ spines, 11 MC–GC pairs, 9 mice). In contrast, the stimulated MC dendrite was stable and did not show any displacement ($0.08 \pm 0.01 \, \mu m$, $n = 7$ MC dendrites, Supplementary Fig. 3). In addition, no differences were observed in the lifetime, directional growth and spine relocation of the SHF when the MC was stimulated with a random pattern (Fig. 4o; Supplementary Fig. 3). These results indicated that spine relocation does not occur passively over time and has to be induced by a physiological pattern of stimulation and not the overall activity of the MC.

**Glutamate released from MC controls the motility of SHF.** Since MC are glutamatergic neurons[24], adult-born GC express both AMPARs and NMDARs[25–27], and since stimulating MC affects the initial stages of GC integration[10] we next investigated whether SHF dynamics and spine relocation depend on MC-derived glutamate and the activation of AMPARs and/or NMDARs on adult-born GC. To address this issue, we extracellularly stimulated the lateral olfactory tract (LOT) where all the axons of MC converge and applied a pattern that mimicked MC responses to odours (Fig. 5a). We used this approach, rather than single-MC stimulation, since it allowed us to acquire prolonged time-lapse images of SHF under BL conditions for 45 min followed by LOT stimulation for 45 min and then to apply NMDAR (APV, 50 μM) or AMPAR (NBQX, 25 μM) antagonists while stimulating the MC for an additional 45-min period (Fig. 5c). To ascertain that the imaged adult-born cell was located in the region activated by the electrical stimulation, we also recorded local-field potentials in that area. Our results showed that MC stimulation stabilizes SHF by increasing their lifetime and reducing their motility (Fig. 5b and Supplementary Fig. 4a–c). These effects depended on the activation of AMPARs, but not NMDARs, since the application of NBQX, but not APV, completely blocked the changes in SHF dynamics and lifetime induced by MC stimulation (Fig. 5b; Supplementary Fig. 4d). No changes were observed under control conditions when we imaged spines of adult-born GC for the same period of time without MC stimulation (Supplementary Fig. 4b, left panel).

While these experiments revealed that AMPARs play a role in the motility of SHF, the extracellular stimulation of thousands of MC cannot be used to assess the directionality of SHF growth or spine relocation. To this end, we applied AMPA locally by iontophoresis and monitored SHF dynamics and spine relocation. We first recorded BL dynamics for 45 min and then positioned a pipette containing 10 mM AMPA and 10 μM Alexa594 (to visualize the position of the pipette) ∼5 μm from the spine of an adult-born GC. A brief pulse of negative current ranging from 150 to 250 nA via the pipette was applied to release the AMPA (Fig. 5d). AMPA iontophoresis increased the lifetime of SHF (Fig. 5e,f) and decreased the dynamics (Supplementary Fig. 4e), which is in agreement with the LOT stimulation (Fig. 5a–c)

and odour administration (Fig. 4a–c) results. Interestingly, however, an analysis of the cosines of the angles between the SHF and the position of the iontophoresis pipette did not reveal any directionality effect by AMPA (Fig. 5g). In addition, we observed no spine displacement following AMPA iontophoresis (Fig. 5h).

**MC-derived BDNF induces spine relocation of adult-born GC.** While our results indicated that glutamate controls the motility of SHF via AMPAR activation, the directional growth of SHF and spine relocation should be regulated by other factors released in an activity-dependent manner. It is well-established that target cell-derived trophic factors may be required for activity-dependent reorganization of synaptic contacts[28,29] and that MCs express BDNF, which affects the spine density of adult-born GC in a TrkB-dependent manner[30]. We thus examined the involvement of MC-derived BDNF in the directional growth of SHF and in spine displacement. We first performed *in situ* BDNF hybridization and observed high levels of BDNF mRNA in the MC and glomerular (GL) layers, but not in the GC layer (Fig. 6a). BDNF fluorescent *in situ* hybridization combined with MC marker PGP9.5 immunolabeling confirmed that MCs expressed BDNF in the OB (Fig. 6b). To address the role of BDNF signalling in SHF dynamics and spine relocation, we pressure-applied BDNF (10 ng ml$^{-1}$) and monitored SHF dynamics on the spines of adult-born GC. Interestingly, puff application of BDNF did not affect the lifetimes of SHF (Fig. 6d,e) but induced directional growth of SHF towards the BDNF-containing pipette (Fig. 6d,f). The directional growth of SHF following the puff application of BDNF was accompanied by the relocation of adult-born GC spines (Fig. 6g,h). Moreover, while we did not observe any relocation of spines with SHF under 45 min of BL imaging, BDNF puff induced relocation of 38.1% of spines with SHF ($n = 21$ spines, 12 cells from 4 mice).

We then produced a BDNF-knockout subpopulation of MC in the adult OB by injecting a Cre-mCherry viral construct above the MC layer of adult BDNF fl/fl mice (Fig. 7a). To ensure the efficacy of the Cre-dependent recombination, we first injected the Cre-mCherry viral construct into GFP reporter mice. We observed multiple GFP+ MC as determined by co-labelling of GFP and mCherry with PGP9.5 (Fig. 7b,c). We then patched the BDNF-lacking Cre-mCherry+ MC in BDNF fl/fl mice, filled it with Alexa594, and monitored the SHF dynamics and spine relocation of GFP+ adult-born GC following a physiological pattern of MC stimulation (Fig. 7d–f). The stimulation of the BDNF-lacking MC still increased the lifetime of SHF (Fig. 7g), which is consistent with a role for MC-derived glutamate in the motility of SHF (Fig. 5). However, no directional growth of SHF (Fig. 7h) and, as such, no spine displacement of adult-born GC (Fig. 7f,i) towards the simulated BDNF-lacking MC was observed. These results suggested that the activity-dependent release of BDNF from MC is required for the directional growth of SHF and the spine relocation of adult-born GC.

**Spines with SHF are maintained after sensory deprivation.** Our results showed that SHF are required for spine relocation and are guided towards MC dendrites by BDNF released in an activity-dependent manner. This structural plasticity may be instrumental in the maintenance of some adult-born GC spines in the constantly remodelling OB network. We thus investigated the SHF dynamics and spine maintenance of adult-born GC in the sensory-deprived OB of BDNF fl/fl mice injected with the Cre-mCherry viral construct (Fig. 8a). We first verified the efficiency of the sensory deprivation by confirming the decrease in TH expression in the GL layer 14 days following nostril

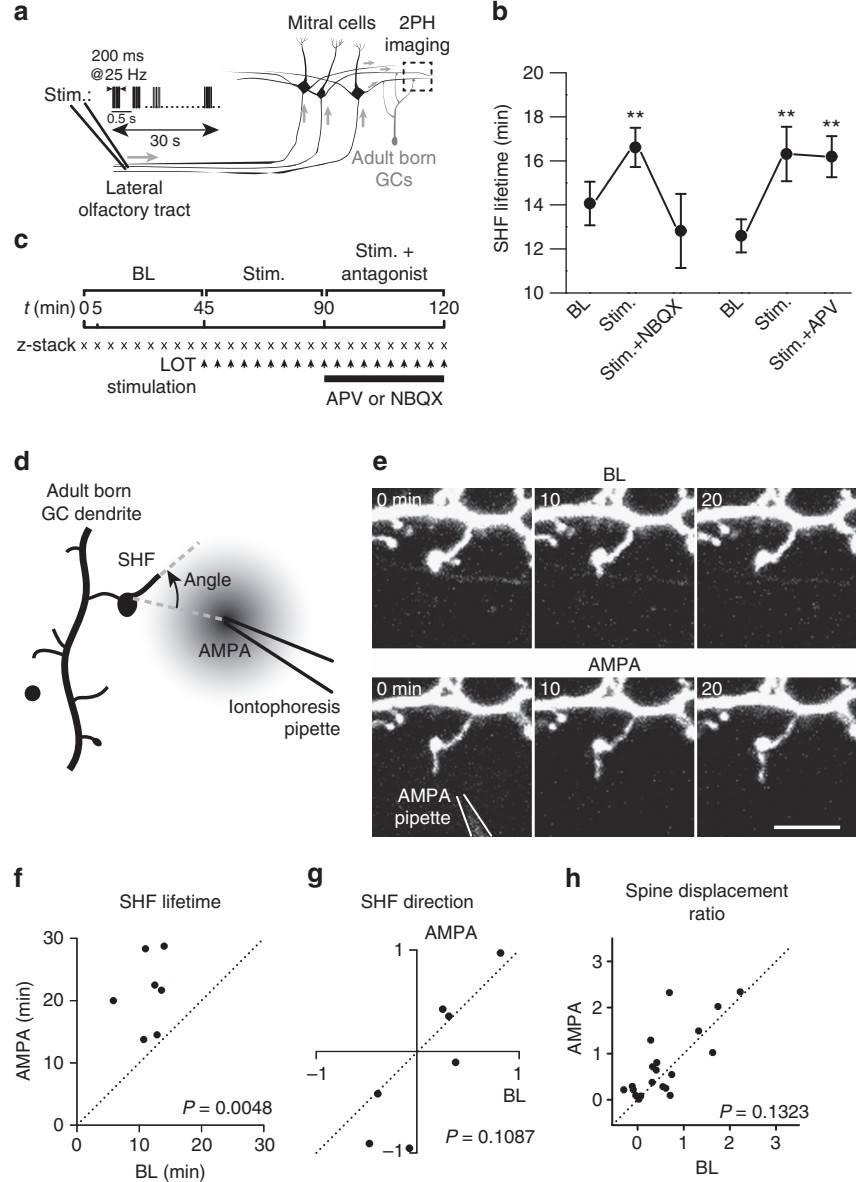

**Figure 5 | Glutamate released from MC stabilizes spine head filopodia. (a)** Illustration of the protocol used to apply a physiological pattern of activity to MC in the OB while imaging adult-born GC. A stimulating pipette was inserted in the lateral olfactory tract (LOT) while an extracellular pipette recorded the local-field potential in the external plexiform layer. A distal dendrite of an adult-born GC was then imaged in the region where MC activity was induced. **(b)** Experimental design of the LOT stimulation experiment. After recording the baseline spine dynamics of an adult-born GC, the dynamics under LOT stimulation were recorded. A glutamate receptor antagonist was applied while maintaining the LOT stimulation. **(c)** Effect of bath application of an AMPA and an NMDA receptor antagonist (NBQX and APV, respectively) on the stabilization of SHF observed after LOT stimulation ($n = 11$ and 12 cells from 7 and 5 mice for NBQX and APV, respectively; $*P < 0.05$ and $**P < 0.01$ using a paired $t$-test). **(d)** Illustration of the protocol used for the local application of AMPA. The angle of the SHF was calculated based on the position of the tip of the iontophoresis pipette. **(e)** Example of an image sequence showing the effect of AMPA iontophoresis on SHF dynamics. Scale bar, 10 μm. **(f)** Quantification of the lifetime of the SHF at baseline (45 min) and following AMPA iontophoresis. **(g)** Quantification of the angle of the SHF with respect to the tip of iontophoresis pipette at baseline and following the application of the AMPA. **(h)** Effect of AMPA iontophoresis on adult-born GC spine displacement over 45 min ($n = 20$ spines). **(f–h)** Data from 7 cells from 7 mice. $P$ values were calculated using the paired Student's $t$-test.

occlusion (Fig. 8b)[31,32]. We then assessed the spine density of adult-born GC in regions containing a high density of Cre-mCherry+ versus Cre-mCherry− MC and compared it with the spine density in sensory-deprived ipsilateral versus control contralateral OBs (Fig. 8c). Sensory deprivation decreased the overall spine density of adult-born GC regardless of the expression of BDNF in MC (Fig. 8d), which is in agreement with previous reports[33,34]. However, an increased percentage of GC spines with SHF in the regions of Cre-mCherry-MC was observed

after sensory deprivation (Fig. 8e). This effect was seen *in vivo*, as well as *in vitro* in acute OB slices derived from sensory-deprived mice (Fig. 8g). These results suggested that sensory deprivation does not affect spines with SHF and leads to the elimination of spines without SHF. Specific maintenance of spines with SHF in the sensory-deprived OB depended on the expression of BDNF by MC since the knockout of BDNF expression by the Cre-mCherry viral construct completely abolished the increase in the percentage of spines with SHF (Fig. 8e, compare quantification

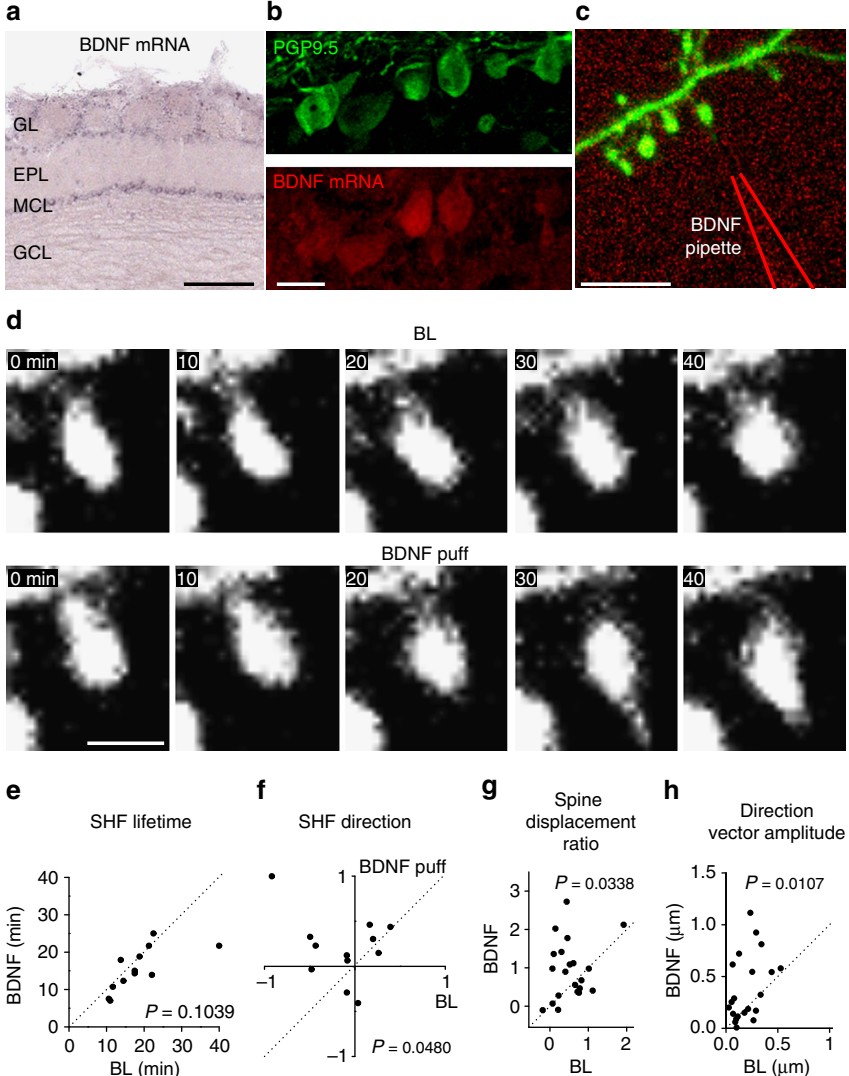

**Figure 6 | BDNF application promotes spine relocation.** (**a**) *In situ* hybridization of BDNF in the OB. Note that BDNF is selectively expressed in the MC and GL layers. EPL, external plexiform layer; GCL, granule cell layer. (**b**) Confocal images of BDNF expression by MC. Green, immunohistochemistry of PGP9.5, a marker for MC. Red, fluorescent *in situ* hybridization of BDNF mRNA. (**c**) Two-photon image of the distal dendrite of an adult-born GC and the location of the BDNF puff pipette. (**d**) Time-lapse acquisition of the spine shown in **c** at baseline followed by the application of BDNF (10 ng ml$^{-1}$) at the beginning of the second imaging session. (**e**) Quantification of the lifetime of the SHF following a BDNF puff. (**f**) Quantification of the directional growth of the SHF towards the BDNF puff. (**g**,**h**) BDNF-triggered spine relocation compared with baseline. Spine relocation was analysed for 45 min under both baseline and BDNF puff conditions ($n = 21$ spines). Normalized spine displacement (**g**) and direction vector amplitude (**h**) are shown. (**e–h**) $n = 12$ cells from 4 mice. $P$ values are calculated using the paired Student's $t$-test. Scale bars: 200 μm, 20 μm, 10 μm and 2 μm for **a–d**, respectively.

for the Cre-mCherry+ and Cre-mCherry− regions in the ipsilateral OB). We next determined whether the presence of spines with SHF in the sensory-deprived OB is due to the increased motility of SHF. We acquired time-lapse two-photon images of adult-born GC dendrites from control and odour-deprived OB to measure the lifetime of SHF. We observed a decrease in the lifetime of SHF and an increase in their dynamics (Fig. 8f). These effects were, however, independent of BDNF expression (Fig. 8f), which is consistent with our observations that MC-derived BDNF affects SHF directional growth and spine relocation but not SHF dynamics (Figs 6 and 7). These results suggested that a reduction in odour-induced activity triggers SHF dynamics in a BDNF-independent manner followed by the specific BDNF-dependent maintenance of spines with SHF.

**Spine relocation is involved in odour information processing.** Previous studies have suggested that inhibition promotes MC

synchronization[35] and that a significant part of this inhibition is conveyed by the GC organized in sparse, segregated and distributed columns[36]. Given these observations, we investigated how and to what extent mature spine relocation affects the synchronization between MCs belonging to different glomeruli and how this structural plasticity impacts odour information processing. To address this issue, we used a three-dimensional (3D) OB network model[37] that represents the natural arrangement of MC and GC and provides a realistic representation of their overlapping dendrites. We started from a network configuration representing three neighbouring GL units after a learning session that resulted in well-formed columns of potentiated GC synapses below each GL (Fig. 9a). At the end of the learning period there were 3,269 GC synapses that were fully potentiated (>95% of the peak conductance value).

Based on our *in vivo* experimental data showing that 3–8% of GC spines relocate (Fig. 1), we ran a simulation in which 3 or 6%

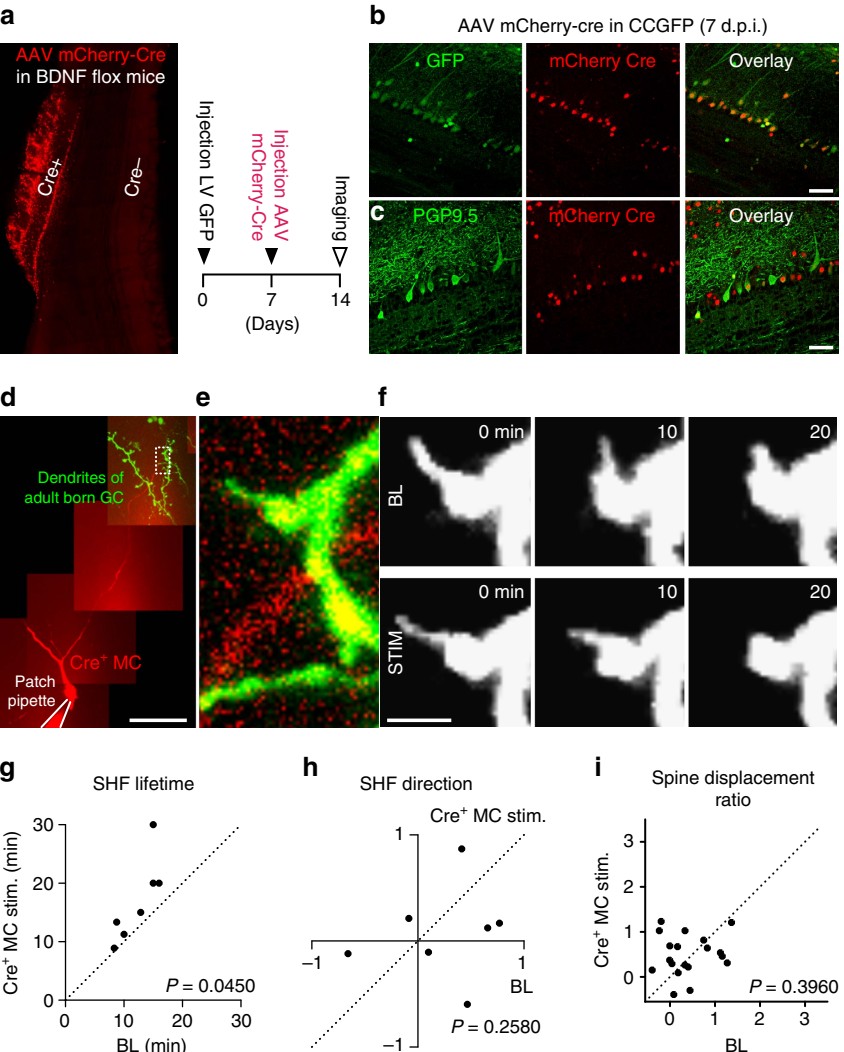

**Figure 7 | The activity of BDNF-lacking MCs does not induce spine relocation.** (**a**) Illustration of the method used to knock out BDNF from a fraction of MCs in the adult OB. GFP-expressing lentiviral particles were first injected into the RMS to label neuronal precursors. Seven days later, an AAV viral vector expressing Cre-mCherry was injected in the MC layer of the OB of BDNF fl/fl mice. The effect of knocking out BDNF was assessed 7 days later (14 d.p.i. for the adult-born GC). (**b**) Confocal images of the GFP signal in the MC layer of the GFP reporter line injected with the Cre-mCherry virus. (**c**) Confocal images of the PGP9.5 protein signal in the MC layer of wild-type mice injected with the Cre-mCherry virus. (**d**) Reconstruction of the lateral dendrites of an MC infected with the Cre-mCherry viral construct and filled with Alexa594 using a patch pipette. (**e**) Enlarged view of the boxed area in **d**. (**f**) Image sequence of a spine illustrating the effect of an electrical stimulation of a BDNF-deficient MC. (**g**) Effect of the activity of a Cre-mCherry + MC on the lifetime of the SHF. These results were obtained from time-lapse images acquired over 30 min of baseline recordings followed by a further 30 min of MC stimulation. (**h**) Comparison of the direction of an SHF forming in baseline conditions and following the stimulation of a BDNF-deficient Cre-mCherry + MC. (**i**) Effect of the stimulation of a BDNF-deficient Cre-mCherry + MC on spine displacement over 30 min (n = 20 spines). (**g–i**) n = 7 cells from six mice. P values were calculated using the paired Student's t-test. Scale bars: 500 μm and 2 μm for **a** and **f**, respectively; and 50 μm for **b–d**.

of the total GC spines that initially connected GL37 with GL86 were switched to connect GL37 with GL123. In general, the number of synapses connecting any two GL units are intrinsically determined by the spatial distribution of the MC and GC dendrites. The synchronization between two GL units was measured as the cross-correlation between the post-stimulus time histograms (PSTHs) of GL37 and of either GL86 or GL123 obtained from 14 sniffs (Fig. 9b, raster plots) using a 20-ms time bin. While a visual inspection of both the raster plots and PSTHs did not reveal any clear difference, the cross-correlation between the involved GL units gradually changed with the proportion of relocated spines from GL86 to GL123 (Fig. 9c). These results suggested that the relocation of relatively few spines in response to a new sensory input can be an effective mechanism for quickly

changing the set of synchronized MC, which in turn affects odour information processing.

## Discussion

We revealed a new form of structural plasticity in the OB that relies on the relocation of the dendritic spines of mature adult-born but not early-born GCs. This relocation is driven by MC activity and is preceded by the growth of SHF from the spine heads. Principal cell-derived glutamate controls the motility of SHF via the activation of AMPARs, whereas activity-dependent BDNF release from MC dendrites drives the directional growth of SHF and spine displacement. Spines with SHF were maintained in sensory-deprived OB, and this maintenance depended on

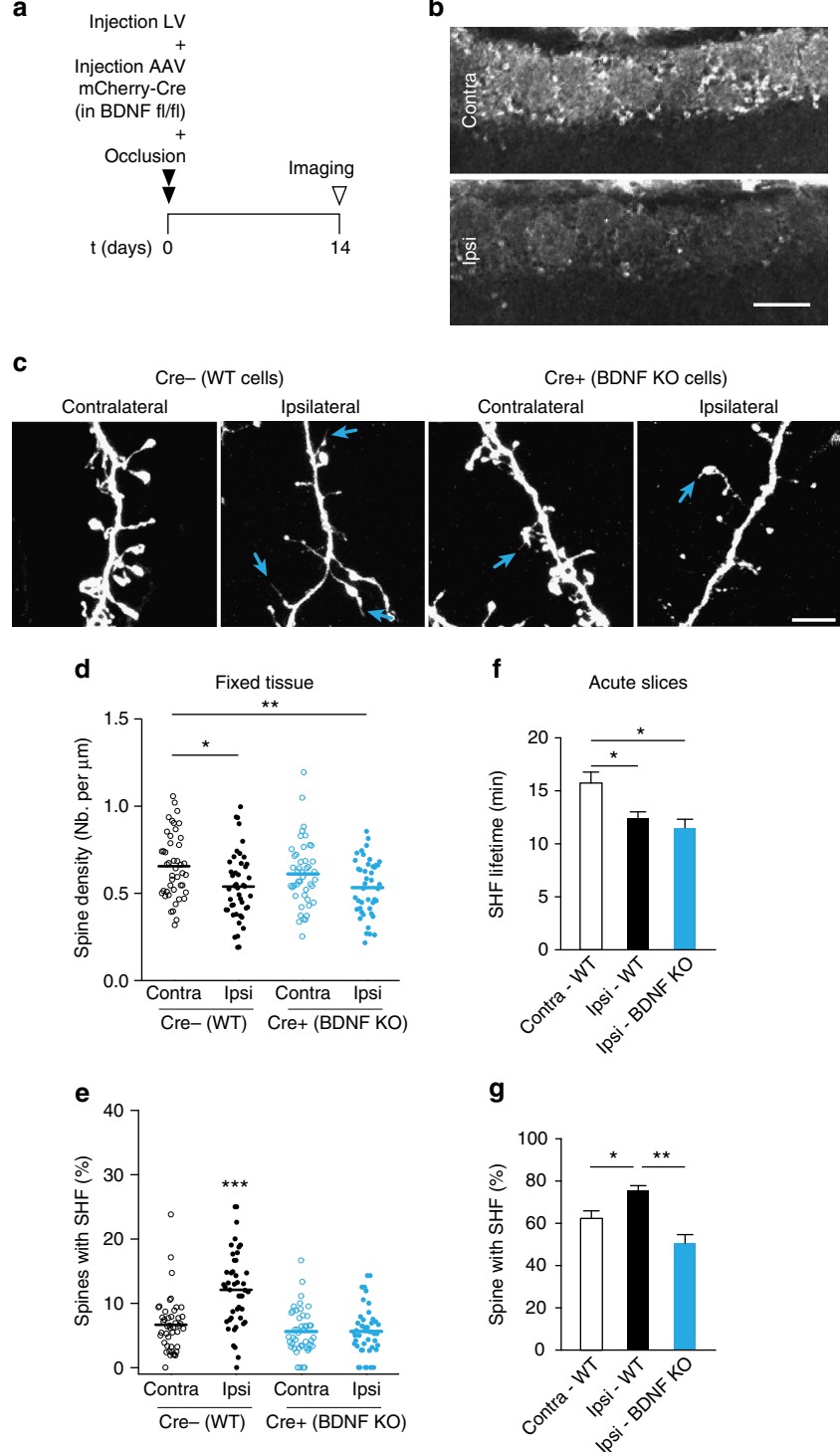

**Figure 8 | Spines with SHF are selectively maintained after sensory deprivation.** (**a**) Experimental design of occlusion and simultaneous injections of GFP and Cre-mCherry viruses. The mice were killed 14 days after the surgery. (**b**) Tyrosine hydroxylase (TH) expression in the glomeruli layer demonstrating the efficiency of unilateral nostril occlusion. Scale bar, 100 μm. (**c**) Representative images of the distal dendrites of adult-born GC in the contralateral and ipsilateral bulbs in injected and uninjected regions of BDNF fl/fl mice. Scale bar, 5 μm. (**d**) Quantification of spine density on adult-born GC in different regions of the OB following sensory deprivation and BDNF knockout. (**e**) Quantification of the percentage of spines with SHF in fixed tissues ($n = 45$ cells from 3 mice in each condition; *$P < 0.05$, **$P < 0.01$, ***$P < 0.001$—one-way ANOVA with Tukey *post hoc* test for **d**,**e**). (**f**) Effect of sensory deprivation on the lifetime of SHF in the control and sensory-deprived bulbs of wild-type mice and in the sensory-deprived bulb of BDNF KO mice. The lifetime of the SHF was measured over a period of 60 min in time-lapse experiments using acute OB slices. (**g**) Percentage of spines with SHF. (**f–g**) $n = 12$ and 9 cells from the contralateral and ipsilateral bulbs of 10 and 6 mice, respectively, and $n = 4$ from 1 BDNF KO mouse. *$P < 0.05$; **$P < 0.01$ using the unpaired Student's *t*-test. ANOVA, analysis of variance

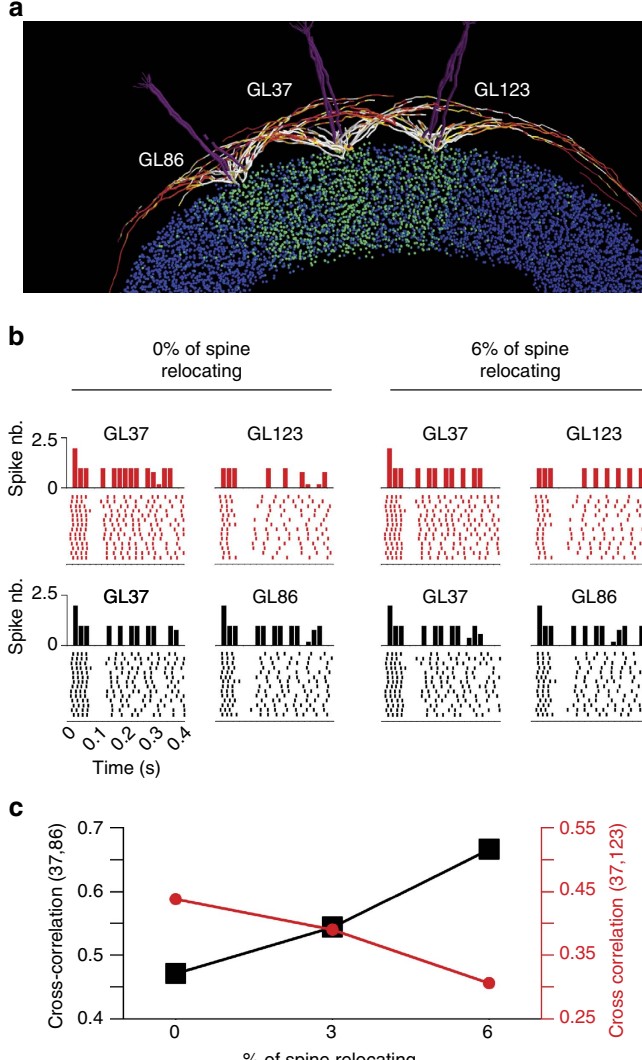

**Figure 9 | Spine relocation promotes fast synchronization of MC with functional consequences for odour information processing.** (**a**) 3D OB network model with glomerular units used in all simulations. Input was delivered to form individual glomerular units. MC dendritic segments are colour coded based on the normalized peak inhibitory conductance they received from the GC. The green coloured points below the MC represent the somas of GCs in which at least one synapse was strongly potentiated (>95% of its peak value). (**b**) Post-stimulus time histograms and the relative raster plot for MC from the different glomeruli under control conditions (left plots) and after spine relocation (right plots). Test simulations were performed by stimulating two glomeruli at a time (GL37 and GL123, red plots) or (GL37 and GL86, black plots). Each simulation lasted for 14 s, with one sniff per second. (**c**) Results of spike cross-correlations between glomeruli as a function of the proportion of relocated spines.

previously suggested that spinules and spine head protrusions may be distinct structures based on differences in microtubule transendocytosis[38]. The filopodia-like structures observed on GC in the OB are long (>2 μm) and resemble, in terms of their length, lifetime and dynamics, the spine head protrusions observed in hippocampal organotypic cultures[40,41] and, *in vivo*, the mouse visual cortex during the period of synaptic plasticity[43]. While both spinules and spine head protrusions are associated with synaptic plasticity[38,39,44], their function is unknown. The results of the present study revealed that SHF are required for a new form of structural plasticity where mature functional spines relocate in an activity-dependent manner. We propose that these SHF may also be required for the structural remodelling of spines in other sensory systems during periods of massive synaptic competition and activity-dependent refinement of neuronal networks.

The release of glutamate and BDNF from MC dendrites has two distinct functions to ensure spine stabilization and spine motility. The release of glutamate from MC may activate the AMPARs on adult-born GC spines, reinforcing the structures that are synaptically active and stopping the formation of SHF in the OB network. It has been suggested that NMDA receptors trigger the formation of new spines[45,46], whereas AMPARs play a role in synapse maintenance by stabilizing actin-based motility[47]. We previously showed that NMDA receptor activity is important for the initial step of spine formation on adult-born GC[10]. We now propose that AMPARs play a role in the stabilization of the spines of adult-born mature interneurons.

On the other hand, the local application of BDNF brought to light a trophic mechanism for inducing SHF directional growth and spine relocation. We hypothesize that, when a period of lower synaptic activity occurs, SHF protrude and grow towards trophic factors such as BDNF released from active MCs. This in turn promotes spine relocation and the synaptic maintenance of adult-born interneurons. The role of BDNF in synaptic maintenance and competition has been studied in the visual cortex where mosaic depletion of BDNF alters spine density[48]. Furthermore, the deletion of TrkB, a high-affinity BDNF receptor, results in fewer spines on adult-born neurons in the OB[30]. We propose that, in the adult OB, the concurrent actions of MC-derived glutamate and BDNF regulate the structural modifications of mature adult-born GC.

Several lines of evidence suggested that SHF guide the relocation of spines from inactive towards active MC dendrites. First, spine relocation was directional and occurred over the space of several micrometres. The MC dendrites were very stable and did not show any displacement, suggesting that GC spines relocate to another stable MC dendrite. Second, the relocating spines were stable and were found at the same position 24–48 h later. Third, the relocated spines expressed pre- and postsynaptic markers and were directly opposed to presynaptic structures, indicating that they are part of dendro-dendritic synapses. These results suggested that relocating spines rapidly form new functional units with their target cells in an activity-dependent manner.

GC-to-MC synapses are involved in the generation of fast evoked oscillations by synchronizing the activity of MC in the OB[3]. Adult-born GCs play an important role in this process by providing ~45% of the inhibition received by MCs[4,34]. These adult-born GCs are highly sensitive to sensory activity[8,49], and odour-induced activity regulates the integration of these neurons into the OB network[33,34]. These reports suggest that the continuous formation, stabilization and/or elimination of new synapses by adult-born GC regulate the functioning of the OB network, which in turn affects some, but not all, olfactory behavioural tasks[50]. The synaptogenesis of adult-born neurons

expression of BDNF by MC. Our modelling studies also suggested that spine relocation contributes to odour information processing and allows for the fast synchronization of MCs in olfactory learning paradigms.

The filopodia-like structures emerging from spines have been observed in primary[38,39] and organotypic[40,41] hippocampal cultures, and in acute slices[42], as well as *in vivo* in the visual cortex during the critical period[43]. The length of these structures ranges from short 167- to 880-nm spinules[39,44] to longer structures called spine head protrusions[40,41,43]. It has been

occurs over time scales that are longer than the rapid and persistent changes in environmental conditions, and the OB would require much faster structural plasticity to react to environmental changes. In our experiments, we did not observe the formation of new spines during 4-h imaging periods of adult-born GC dendrites. This contrasts with spine relocation, which can occur within a few minutes. In this respect, the relocation of mature spines of adult-born GC from inactive towards active MC dendrites may allow a much faster adaptation of the OB network to rapidly changing environmental conditions. Indeed, our computational modelling experiments revealed that the relocation of a few spines may be sufficient to synchronize a new subset of principal cells during new sensory input processing. Our study revealed a new form of activity-dependent structural plasticity that allows for the rapid adaptation of the OB network to new sensory inputs.

## Methods

**Animals.** Adult ($>$2-month-old) male C57BL/6 mice (Charles River), postnatal day 10 (P10) C57Bl/6 and CD1 mice (Charles River), adult male and female BDNF fl/fl mice (Jackson Laboratories), and CAG-CAT-EGFP reporter mice (kindly provided by Dr Kenneth Campbell, Cincinnati Children's Hospital Medical Center) were used. The experiments were performed in accordance with Canadian Guide for the Care and Use of Laboratory Animals guidelines. All the animal procedures were approved by the Université Laval animal protection committee. One to four mice per cage were kept on a 12-h light/dark cycle at a constant temperature (22 °C) with food and water ad libitum.

**Stereotaxic injection.** A GFP-encoding lentivirus (100–300 nl; $1 \times 10^9$ i.u. ml$^{-1}$, UNC Vector Core) was injected into the RMS of both brain hemispheres of the mice to label adult-born GC. The following coordinates were used for 20–25 g mice (from bregma): anterior–posterior: 2.55 mm; medial–lateral: $\pm$ 0.82 mm; and dorsal–ventral: 3.15 mm. After injection, the mice were returned to their cages and were kept for different periods of time (14, 28, 42 and >77 d.p.i.). The >77 d.p.i. group included animals kept for up to 163 d.p.i. We detected few GFP+ cells in the RMS of the OB, indicating that the viral construct did not infect stem cells[10]. With our injection method, we thus obtained an accurate assessment of the age of the adult-born neurons. In some experiments, we infected neuronal progenitors in the RMS with td-tomato-encoding lentivirus particles (100–300 nl; $1.5 \times 10^{10}$ i.u. ml$^{-1}$, UNC Vector Core) to perform Ca$^{2+}$ imaging of adult-born GCs spines with OGB. To label early-born neurons, we injected a GFP-encoding lentivirus into the RMS of P10 mice at the following coordinates (from bregma): anterior–posterior: 2.05 mm; medial–lateral: $\pm$ 0.65 mm; and dorsal–ventral: 2.7 mm. To selectively knockout BDNF expression in the OB, an AAV Cre-mCherry ($2 \times 10^{12}$ i.u. ml$^{-1}$, UNC Vector Core) viral construct was injected into the OB of BDNF fl/fl mice. The following coordinates were used: anterior–posterior: 5.00 mm; medial–lateral: $\pm$ 1.50 mm; and dorsal–ventral: 1.38 mm.

**Time-lapse two-photon imaging *in vivo*.** Adult mice injected with a GFP-encoding lentivirus in the RMS either at P10 or P60 were used for in vivo two-photon imaging at 45–60 d.p.i. (for P10 mice) or 30–50 and 120–150 d.p.i. (for P60 mice). The mice were anesthetized with 2–3% isoflurane during cranial window implantation. The temperature was maintained at 37.5 °C during the entire procedure using an infrared blanket (Kent Scientific). After removing the scalp, we drilled a circular craniotomy centred over the OB hemispheres. We removed the bones protecting the OB, leaving the dura intact. Bleeding was controlled with gel foam pieces. A 3-mm-diameter coverslip was centred on top of the OB, and Kwik-seal (World Precision Instruments) and dental cement was used to maintain the coverslip on the surface of the OB. A small head plate glued to the skull was used to keep the head from moving during the imaging procedure. Once the surgery was completed, the amount of isoflurane was gradually decreased, and the mouse was injected with ketamine-xylazine to maintain anesthesia during the imaging period.

GC structural dynamics were imaged using a custom-made video-rate two-photon microscope[51]. Thirty to 60 min after surgery, we positioned the mouse using the head plate on a custom-made stereotaxic frame controlled by a micromanipulator (MPC 200; Sutter). A $\times$ 20 water-immersion objective (XLUMPlanFl $\times$20/numerical aperture (NA) 0.95; Olympus) was used to locate the region-of-interest for dendritic imaging. For some isolated GFP+ GC, we used a simple neurite tracer plugin to reconstruct the morphology of the neuron (ImageJ). Structural dynamics were imaged using a $\times$ 60 water-immersion objective (XLUMPlanFl/IR $\times$ 60/NA 0.90). Seventy- to 100-μm image stacks (2 μm steps) were acquired in the external plexiform layer at a rate of one image stack every 5 min. At the beginning of each session, GCs were identified by locating their soma,

which were 200–500 μm down from the brain surface. For chronic imaging experiments, GC spines were first imaged on day 1 for 2–4 h and were re-imaged 24–48 h later.

To study the effects of sensory activity on the structural dynamics of GC, we delivered odours with an olfactometer (Knosis) at the beginning of each stack acquisition (30 s of odour delivery every 5 min). We used a mixture of butyraldehyde and methylbenzoate at a concentration of $10^{-3}$ that has been shown to activate the medio-lateral region of the dorsal surface of the OB[22]. The effect of the sensory activation was compared with the results of a 60-min BL period in the absence of odour.

**Time-lapse two-photon imaging *in vitro*.** Deeply anesthetized mice were transcardially perfused with ice-cold oxygenated artificial cerebro-spinal fluid (aCSF–sucrose) containing the following (in mM): 250 sucrose, 3 KCl, 0.5 CaCl$_2$, 3 MgCl$_2$, 25 NaHCO$_3$, 1.25 NaH$_2$PO$_4$ and 10 glucose. Horizontal slices (250–300-μm-thick) of the OB were obtained using a vibrating blade microtome (HM 650 V; Thermo Scientific). The slices were transferred into oxygenated aCSF maintained at 32 °C that contained the following (in mM): 124 NaCl, 3 KCl, 2 CaCl$_2$, 1.3 MgCl$_2$, 25 NaHCO$_3$, 1.25 NaH$_2$PO$_4$ and 10 glucose. For the time-lapse two-photon imaging experiments, we placed the acute slices in a perfusion chamber equipped with a temperature controller (Warner Instruments). We acquired images of the spines of apical dendrites of GFP-expressing GCs located at least 40-μm-deep in the slices using a custom-made two-photon microscope (modified from an Olympus FV 300 system) equipped with a $\times$ 60 water-immersion objective optimized for infrared light imaging (LUMPlanFl/IR $\times$ 60/ NA: 0.90; Olympus) and a Ti–Sapphire laser (Coherent). To assess spine dynamics, time-lapse z-stack images were acquired every 5 min using Fluoview 5.0 software (Olympus). Two detection channels allowed us to simultaneously image the GFP and Alexa594 signals for the MC/GC and puff/iontophoresis experiments. GFP and Alexa594 were excited at 900–940 nm. The same imaging system was used for Ca$^{2+}$ imaging in adult-born GC spines. For these experiments, adult-born GCs were labelled with td-tomato-expressing lentiviral particles and were filled with OGB organic Ca$^{2+}$ indicator (100 μM) at 21–28 d.p.i. via a patch electrode. The Td-tomato-expressing GCs were excited with a 1,040 wavelength laser (Spectra-Physics). Ten to 15 min after filling the GC with OGB, line scan imaging was performed on spines with and without SHF following a single depolarizing pulse (200 ms) applied using a patch electrode to induce action potential.

**Image analysis.** The time-lapse images were analysed using custom-made programmes written in MATLAB (version R2010a; Mathworks). We extracted maximum projection 512 $\times$ 512 pixel images of each time point. The slice drift between each time point was corrected using a cross-correlation-based algorithm. The original images were cropped to facilitate the visualization and analysis of structural dynamics. For some analyses, we also false-coloured every time point (previous stack, red; current stack, green) to detect appearing and retracting SHF, as well as spine displacement.

Several criteria must be met to detect SHF. First, they have to clearly protrude out of the spine head by at least 1 μm. Second, they need to be easily identifiable with a high-signal-to-noise ratio. Third, they must protrude from a spine identified as stubby or mushroom, that is, the spine head volume has to be considerable. Fourth, we analysed SHF on fairly isolated spines while spines with SHF in 'crowded' regions were not analysed. Last, they have to be transient. SHF that were present during the whole time course of a recording were discarded. For example, SHF that appeared during BL conditions and were maintained during drug or MC stimulation conditions were not quantified. We only took the spines that appeared during one block of the experiment (either BL, STIM or Drug + STIM) into consideration. We used these criteria to quantify SHF dynamics on adult-born GC by calculating the number of SHF forming/retracting on spines for a period of 1 h per μm of dendrite. The number of SHF was obtained by manually counting the number of SHF forming or retracting on adult-born GC spines. The SHF lifetimes were calculated by tracking each SHF manually using the MTrackJ plugin (developed by Dr Erik Meijering, Biomedical Imaging Group, the University Medical Center Rotterdam, Rotterdam, The Netherlands) for ImageJ. For all quantifications involving SHF dynamics and lifetime, data were averaged per cell to avoid any 'spine' bias. The resolution of the lifetime measurements was limited by the sampling rate (5 min) and the length of the time-lapse acquisition (30–240 min, depending on the experiment). SHF tracking also made it possible to measure the direction of SHF growth. The angle of SHF growth was measured based on the position of either the MC lateral dendrite or the location of the puff/iontophoresis pipette (see illustration in Fig. 4j). We averaged the direction angle for each time point during the SHF observation period and calculated the cosine of this angle to obtain its direction. The direction measurement ranged from +1 (an SHF pointed directly at the target) to −1 (an SHF pointed away from the target).

We assessed spine motility and volume using a custom-made programme to automatically detect the contour of the spine head. In these experiments, the investigator was not blinded to the experimental condition since, in many cases, spine relocation was measured under BL conditions with no experimental manipulation (Figs 1–3). To avoid any bias in the analysis of spine relocation, we automatically tracked a large population of randomly selected spines (514 spines) and used statistical methods such as z-score of direction vector

amplitude (Fig. 1i), allowing for stringent, statistical definition of spine relocation. Spine relocation was thus defined by automatically set parameters based on statistical measures and not by the experimenter. For the experiments revealing the molecular mechanisms of spine relocation, the same spines were tracked under BL and experimental conditions and, once again, statistical values were used to define spine relocation. This approach was based on quantitative automatic measurements of spine movement using a preset threshold value and not on a qualitative assessment of the number of spines that relocate. The preset threshold value was consistent across all conditions. Spine relocation was measured based on the centre of the spine contour for each time point. These values were normalized to those of the region of the dendrite from which the spine emerged. The spine displacement values were determined for 30–90 min, depending on the experiment. We also calculated the direction vector to assess the directionality of spine relocation. The direction vector was calculated by averaging the vector between the origin ($t = 0$ min) and each subsequent time point. For analysing spine head relocation, we used spines that had a good GFP signal-to-noise ratio throughout the whole imaging session and avoided spines that were clustered together to facilitate tracking. We also investigated the spatial distribution of spine relocation and SHF plasticity by measuring the relationship between the number of SHF per spine and the distance of each spine from either relocating or non-relocating spines. The distance was measured by drawing a straight line between the two spines.

**Stimulation of MCs.** To assess the role of MC activity on the motility of spines of adult-born GC, we stimulated the LOT where all the axons of MC of the OB converge[10]. Briefly, the LOT was electrically stimulated using 0.1-ms pulses at $\sim 100\,\mu A$ using an A360 stimulus isolator (World Precision Instruments) triggered by a Digidata 1440 A data acquisition system (Molecular Devices). The pattern of electrical stimulations (five 0.1-ms pulses at 25 Hz repeated 60 times every 500 ms) was designed to reproduce the typical response of MC to odours[23]. We assessed the specificity of the specific stimulation pattern by comparing the results with a random stimulation pattern in which the same number of spines were evoked after random delays. We recorded the local-field potential responses in the EPL using a Multiclamp 700B amplifier (Molecular Devices). We positioned the stimulating pipette as far as possible from the imaging region to avoid direct stimulation of the adult-born GCs. To investigate the role of NMDA and AMPARs in the SHF dynamics of adult-born GCs, we bath-applied 50 μM APV (NMDA receptor antagonist) or 25 μM NBQX (AMPAR antagonist) during the LOT stimulation protocol.

To assess the role of a single-MC stimulation on the spine dynamics of adult-born GC, we performed whole-cell patch-clamp recordings of MC. Electrophysiological recordings and MC stimulations were performed with a Multiclamp 700B amplifier. Patch electrodes (ranging from 2.5 to 4 MΩ) were filled with an intracellular solution containing (in mM): 122.5 K-methylsulfate, 10 KCl, 10 HEPES, 0.2 EGTA, 2 ATP, 0.3 GTP and 10 glucose. We added 10 μM of Alexa594 fluorescent dye (Life Technologies) to the intracellular solution to visualize MC lateral dendrites. The MC was maintained in current-clamp mode during the recording. We recorded time-lapse sequences of adult-born GC spines located within 5 μm of the lateral dendrite of the MC. A BL period of 30 min in which no current was applied to the MC was followed by a stimulation period in which the current was applied to the MC during each acquisition. The pattern of the injected current was as follow: 150 ms pulses of $\sim 100\,pA$ repeated every 500 ms 60 times. MC with a resting potential above –50 mV were discarded. We used spines located at least 10 μm away from the stimulated MC dendrite as controls. In the experiments investigating the role of MC-derived BDNF in spine motility, we took recordings from MC infected with the Cre-mCherry viral construct. The construct was injected 7 or 14 days before the experiment. We clearly identified the Cre-mCherry+ cells by bright field and fluorescence signals before approaching the patch pipette. Cre-mCherry− cells located far from the injection site were used as controls.

**Iontophoresis and puff application.** We used iontophoresis to locally apply AMPA to the spines of adult-born GC distal dendrites and to assess changes in SHF dynamics and spine displacement. After recording BL dynamics for 45 min, we placed a pipette filled with AMPA (10 mM diluted in aCSF) 5 μm from the spine of interest. We applied one 2-ms pulse of negative current (ranging from 150 to 250 nA) at the beginning of the second period of time-lapse acquisition using an ION-100 apparatus (Dagan). To assess the trophic role of BDNF in SHF formation on spines, we pressure-applied BDNF (10 ng ml$^{-1}$) for 10 ms at 10 psi at every time-lapse acquisition using a PV930 Pneumatic PicoPump (World Precision Instruments). In both the puff and iontophoresis experiments, the position of the pipette was visualized by adding Alexa594 (10 μM) to the solution. SHF dynamics were calculated for a $20 \times 20\,\mu m$ region centred on the pipette. For the iontophoresis application experiments, we also calculated SHF dynamics at remote positions on dendritic segments of adult-born GCs located >10 μm from the pipette. We used the tip of the pipette in the puff and iontophoresis application experiments as the reference for measuring the direction angle of SHF. The pressure application of aCSF was used as a control.

**Unilateral nostril occlusion.** The mice were anesthetized with an injection of ketamine-xylazine. Their reflexes were monitored to assess the depth of anesthesia

before beginning the surgery. Occlusion plugs were fabricated using polyethylene tubing (PE50, I.D. 0.58 mm, O.D. 0.965 mm; Becton Dickinson), with the centre blocked using a tight fitting knot made from Vicryl sutures (3-0; Johnson & Johnson). The $\sim 5$-mm-long petroleum jelly-coated plugs were inserted in the left nostrils of the mice ($n = 12$). We verified that the tip of the plug was fully inserted in the nostril to avoid removal by the mouse. The nostrils were occluded before stereotaxically injecting the GFP lentivirus in the RMS and the AAV Cre-mCherry viral construct in the OB. We assessed the success of the occlusion 14 days later by comparing the TH-labelling intensities of the ipsilateral versus the contralateral bulb. Only mice showing a significant decrease in TH-labelling intensity were used in the study.

**Immunohistochemistry.** The mice were given an overdose of pentobarbital and were perfused transcardially with 0.9% NaCl followed by 4% PFA. The OB were resected and were post-fixed in 4% PFA at 4 °C. Horizontal sections (50-μm-thick) were cut using a vibratome and were incubated with the following primary antibodies: mouse anti-TH (24 h, 1:1,000, catalogue number 22941; ImmunoStar), rabbit anti-PSD 95 (48 h, 1:1,000, catalogue number 51-6900; Invitrogen), rabbit anti-synaptoporin (48 h, 1:500, catalogue number 102 002, Synaptic System), rabbit anti-GFP (24 h, 1:1,000, catalogue number A11122; Molecular Probes), rabbit anti-PGP 9.5 (24 h, 1:1,000, catalogue number 31 A; Ultra Clone), mouse anti-Cre (24 h, 1:1,000, catalogue number MAB3120; Millipore) and anti-mCherry (24 h, 1:1,000, catalogue number 5411-100; Biovision). The corresponding secondary antibodies were used. The immunohistochemistry for PSD95 and synaptoporin was performed on 4 mice, for GFP, PGP9.5 and Cre on 4 mice, for TH on 12 mice and for mCherry on 3 mice, respectively. Six animals used for *in vivo* imaging were processed for anti-synaptophysin labelling. Anti-synaptophysin antibody was coupled to a fluorescent antibody using Mix-n-Stain CF568 antibody labelling kits (Biotium). Anti-synaptophysin immunohistochemistry was performed on 100-μm-thick OB sections. The sections were incubated for 72 h at 4 °C. To find the same GC dendrites in the OB slices, the mice used for the *in vivo* imaging were perfused. The post-fixed brains were placed in 4% agar and were positioned on the set-up used for the *in vivo* imaging. The same dendrites were found, and a high power Ti–Sapphire laser beam was directed at the same depth as the imaged spines but 300–500 μm apart in the y or x-axis to burn small regions. The OB was then cut into 100-μm-thick sections. The section containing the burned region, as well as two adjacent sections were kept for immunohistochemistry. Fluorescent images of fixed tissues were obtained using a confocal microscope (FV 1000; Olympus) with $\times 100$ (UPlanSApoN $\times 100$/NA 1.40; Olympus) and $\times 40$ (UPlanSApoN $\times 40$/NA 0.90) objectives.

***In situ* hybridization.** Antisense RNA probes were labelled using DIG RNA labelling kits (Roche Diagnostics) and were purified on ProbeQuant G-50 columns (GE Healthcare). *In situ* hybridization was performed on 50-μm-thick vibratome sections from five mice. The signals were revealed with nitroblue-tetrazolium-chloride/5-bromo-4-chlor-indolyl-phosphate (Promega). The antisense probes were obtained from plasmid-containing mouse BDNF (kindly provided by Dr Castren, the University of Helsinki, Finland). *In situ* hybridization was performed on 40-μm-thick free-floating vibratome sections. The sections were prehybridized in hybridization solution (50% formamide, 4% SSC, 0.05% Denhardt, 0.05% salmon sperm DNA, 0.025% tRNA) for 1 h at 45 °C and were hybridized overnight with a Dig-labelled BDNF antisense probe. The sections were then incubated with an anti-Dig antibody (Roche Diagnostics) overnight at 4 °C. The signals were revealed using nitroblue-tetrazolium-chloride/5-bromo-4-chloro-indolyl-phosphate. BDNF mRNA images were obtained using a bright field microscope (BX51; Olympus) equipped with a $\times 20$ objective (UPlanSApo $\times 20$). For fluorescent *in situ* hybridization, an anti-Dig-POD antibody was used, and the signal was amplified with the TSH-Plus tetramethylrhodamine system (PerkinElmer) using the manufacturer's protocol.

**OB network model.** We used a recent novel 3D model of the OB[37] that represents the natural 3D arrangement of MC and GC, including their overlapping dendrites, for all the simulations. Briefly, the model implemented the reported spatial distribution of 127 glomeruli distributed over $\approx 2\,mm^2$ of the dorsal area and activated by natural odours. The model was composed of 635 MCs (5 for each glomerulus) and 97,017 GCs, which is consistent with the commonly accepted estimate for the MC/GC ratio. We selected a subset of 15 MCs projecting to 3 different glomeruli (GL37, GL86 and GL123) and the relative ensemble of GC ($n = 14652$) connected to them through dendro-dendritic synapses. Unless noted otherwise, we will refer to the ensemble of MC projecting to a given glomerulus and the set of GC with synaptic connections to them as a glomerular unit.

The odour input and synaptic plasticity rule were identical to those used in Migliore et al.[37]. Briefly, all the synaptic weights started at zero and, in response to an odour input, each component (inhibitory or excitatory) of each dendro-dendritic synapse was independently modified according to the local spiking activity in the lateral dendrite of the MC or in the GC synapse. The characteristics of the synaptic clusters predicted by this model were consistent with those observed experimentally, and their formation was an extremely robust process. Unless noted otherwise, a learning session consisted of a 7-s odour stimulation, which was

sufficient to achieve a stable configuration of the synaptic weights under all conditions.

All simulations were carried out in a fully integrated NEURON + Python parallel environment (NEURON v7.3)[52] on a BlueGene/Q IBM supercomputer (CINECA, Bologna, Italy).

**Statistical analysis.** The data are expressed as means ± s.e.m. for bar graphs, individual values for scatter plots, and individual values and means for population comparisons. The normality of the samples was assessed using the Lilliefors test. Statistical significance was tested using a paired or unpaired two-sided Student's $t$-test, depending on the experiment. Equality of variance for unpaired $t$-test was verified using F-test. One-way analysis of variance and Tukey *post hoc* tests were used to compare the groups at different times after injection. The levels of significance were set as follows: $*P < 0.05$, $**P < 0.01$ and $***P < 0.001$.

**Data availability.** The data that support the findings of this study are available from the corresponding author upon request. The model and simulation files used for the present work are available for public download in the ModelDB section of the Senselab database suite (http://senselab.med.yale.edu, accesion number 186771).

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

## Acknowledgements

We thank Drs Kenneth Campbell (Cincinnati Children's Hospital Medical Center) for providing the CAG-CAT-EGFP reporter mice, E. Castren (the University of Helsinki, Finland) for providing the BDNF plasmid, and Gordon M. Shepherd for useful discussions. We also thank Mireille Massouh for preparing illustration for the cover art. This work was funded by an operating grant from the National Science and Engineering Research Council of Canada (NSERC) and a grant from the Canadian Institutes of Health Research (CIHR) to A.S. V.B.-P. was supported by a PhD fellowship from FRSQ and a training grant in neurophotonics from CIHR. M.M. and F.C. are also grateful for support of the SenseLab project by grant 01 DC 00997701-06 from the National Institute of Deafness and Other Communication Disorders, the CINECA consortium (Bologna, Italy) and the PRACE association (Partnership for Advanced Computing in Europe) for granting access to the IBM BlueGene/Q FERMI system. D.C. holds a Canada Research Chair in biophotonics, and A.S. holds a Canada Research Chair in postnatal neurogenesis.

## Author contributions

A.S. and V.B.-P. designed the study. V.B.-P. performed most of the experiments and analysed the data. K.B., V.B.-P. and A.S. performed the *in vivo* imaging experiments. D.H. examined the spine dynamics of adult-born neurons in the control and odour-deprived olfactory bulb and the random stimulation experiments. R.R.B. performed the $Ca^{2+}$ imaging experiments. M.S. performed the immunohistochemistry and *in situ* hybridization. D.C. provided the video-rate two-photon system for the *in vivo* imaging. M.M. and F.C. performed the modelling studies. M.M. compiled and drafted the modelling results. A.S. supervised the project and obtained the funding. V.B.-P. and A.S. wrote the paper.

## Additional information

**Competing financial interests:** The authors declare no competing financial interests.

