## [Peer review file · Nature Communications]

Transferred manuscripts:

Editorial Note: this manuscript has been previously reviewed at another journal that is not operating a transparent peer review scheme. This document only contains reviewer comments and rebuttal letters for versions considered at Nature Communications. Reviewer #1 also reviewed for the other journal while we recruited a new reviewer, Reviewer #4 to provide comments on the revised manuscript.

Reviewers' Comments:

Reviewer #1 (Remarks to the Author)

The authors have done a good job in revising their manuscript. I was critical of an essential part of the data set and of the documentation embedded in Figure 3. In the revised version of the paper, new experiments have been included and a scrutiny of these convinces me that the original conclusions are robust and valid. It is now clear that odor stimulation stabilizes SHF of adult born GC at both 30-50 and 120-150 dpi. It adds to the quality of the paper that the figure legends have been revised for clarity and accuracy.

In the original version of the manuscript I found the text to be overlong and with redundancies that detracted from the quality of the study. Again, improvements have been made. As it now stands, the manuscript reads well, with a few exceptions detailed below. Importantly, the abstract has been revised so as to provide a more accurate summary of the main findings.

There are some outstanding issues that must be dealt with:

1. Even though the abstract has been revised and improved, it stills falls short of conveying effectively to the reader the novelty of the present paper. As I see it, the most interesting finding relates to time scales. As referred to in the Introduction, some of the environmentally induced changes in OB function are known to be rapid while the synaptogenesis of adult-born neurons occurs over a longer time scale. This discrepancy in time scales implies that there must be structural modifications at play that are quicker than those that depend on the development of new synapses. The abstract addresses this issue in a rather oblique fashion. Some additional modifications are required.
2. The abstract states that there is an "unusual form of structural remodeling". The term "unusual" strikes me as odd, even more so since the authors (in the discussion) speculates that this type of spine relocation may occur also in other neuronal networks and thus be a phenomenon that is not restricted to the olfactory system.
3. The text needs to be checked carefully for misprints and inconsistencies. Misspellings abound. And there are some unorthodox terms and expressions - like "synaptophysin+" on p. 4, "thrilling to propose" on p 19, and "all this suggest" on p 20. Clearly the manuscript must be subjected to a final check before resubmission.

Reviewer #4 (Remarks to the Author)

The authors observe motile spine head filopodia (SHF) in the granule cell dendrites in the olfactory bulb. They show that a larger total number of SHF (imaged over 4 hours) correlate with greater normalized spine relocation(aka displacement) and greater amplitude of the direction vector of displacement (microns) in the olfactory bulb. These SHF bearing spines are more common and more motile on adult born compared to early born granule cells. Spine relocation is seen as a method for rapid structural plasticity.

To examine the mechanism by which spine head motility direction can be related to spine relocation direction, the authors show olfactory stimulation enhances motility (linking it to

activity). In acute slices they then find patterned but not random mitral cell activity alters SHF and relocation. Spine head filopodia motility is dependent on AMPA and direction can be related to BDNF. Occlusion and BDNF mitral cell KO interact to alter spine density and % of spines with SHFs. A model is used to suggest % changes observed in vivo could alter olfactory encoding.

This paper contains a wealth of information about mechanisms but could be more clear about the basic finding, what it is and is not, and its relative occurrence compared to other forms of structural plasticity. Namely what is the motility of spines without SHF, spine gains and losses (with and without SHF), and changes in spine volume (with and without SHF). When there is a spine with a SHF is this a hotspot generally for the whole dendrite?

1. What is (and is not) relocation? Also displacement (a term used for figures)? The manuscript would greatly benefit from a more clear definition of spine relocation and the criteria for scoring (for inclusion and exclusion). Also how can relocation be distinguished from a situation where a spine is lost and a new one grows nearby or on the other side of a dendrite? How does it differ from spine neck motility in general?

Fig. 1d lower row. It is not clear to me that this is an example of relocated spine rather than a new spine (the authors claim to not have ever observed a new spine). Here near the initial arrow there is still a structure remaining on the lower side and to my eye there may be gains on the upper side. It is also unclear if this is the same spine on Fig 3d lower panel on the first day and last (with a gap of imaging in time which cannot be accounted for). As a reader I'd like to know what percentage of relocation events fall into this more ambiguous category? The more ambiguous cases might muddy the data for more conservatively defined events. The methods do not adequately address the subjectivity of the confusion between relocation and new growth for these examples. Cases that show translocation across the dendrite or in which new neighboring spines might also be relocated originals should be excluded to eliminate these ambiguities.

2. What happens in the local and larger neighborhood of spines with SHF? When spines relocate or have SHF is there also higher nearby hotspot activity compared to the rest of the dendrite? There seem to be small spines in many of the images that are transient or part of a compound that become discriminable. This is despite methods suggesting crowded regions were avoided.

3. What does the motility of spines that do not have SHF look like when compared to spines that do? Do non SHF spines relocate too? This is perhaps captured by Fig 1i but could be discussed more clearly and could be explicitly compared to spines with SHF.

4. What spine volume changes (measured by intensity normalized to backbone) occur with relocation? Do they dim before relocation or do they always grow? Also does this occur to neighbors?

5. Writing and interpretation of causality is an issue.

a) The abstract states: "This relocation is induced by the growth of spine head filopodia (SHF) and is driven by principal cells activity." The interpretation here could be toned down. Also replace "preexisting" GCs with more specific descriptor like early born.

b) Discussion of Fig 8 (in results and discussion) should consider alternate orders of causal events. Especially if SHF are by definition transient to the time of the experiment, they may simply tend to be present later on spines that survive, possible spines close to a source of BDNF, and SHF themselves may not play a causal role in spine survival.

c) Language in the discussion of the model also suggest experiments were actually done "We also show that spine relocation contributes to odor information processing and allows for fast synchronization of MC in olfactory learning paradigms."

6. When showing "n" this is often SHF or spines. Please state which and also indicate in how many individual cells and mice.

7. Panel 1I does not fully address Reviewer 2 question 4. This should also be done on spines with SHF to address this question: "How much of the SHFs turned into spine head relocation in various

experimental conditions examined (Fig. 1-7)."

8. Fig 3 addresses reviewer #3 request but should also compare to non relocating spines.

9. Reviewer #3 also asked about nearby vs. distal spines and the impact of stimulation (Is the stimulation generally potent to elicit changes near stimulation, or it more global to the connected neuron? Actually, what is the distance of effective activation here?) In the rebuttal the authors point to a figure on SHF lifetime (4i), to suggest the effect on spine relocation is local. Similar data showing spine relocation would make this point.

10. New figure 3 images show relocating spines but these do not have SHF (or they are not clear). Were they present in other images? Also same issue for 4h. Also Fig 3, the images showing re-identification of a dendrite in fixed tissue is also not convincing. The spines do not match up. This also begs what data connects relocation exclusively to SHF possession? This is relevant to Reviewer 2 question 4 above in item 7.

11. Spine relocation (or displacement as it is called here) is likely to grow with time. So it seems a priori that plots like 4o showing a relocation bias in the second period are also likely to show a bias under control conditions. However random stimulation experiments added in this round don't find this. The control experiment plot should perhaps be considered for the main manuscript and the discrete timing of relocation could be discussed.

12. Was the data scored blind to condition?

Referee #1

The authors have done a good job in revising their manuscript. I was critical of an essential part of the data set and of the documentation embedded in Figure 3. In the revised version of the paper, new experiments have been included and a scrutiny of these convinces me that the original conclusions are robust and valid. It is now clear that odor stimulation stabilizes SHF of adult born GC at both 30-50 and 120-150 dpi. It adds to the quality of the paper that the figure legends have been revised for clarity and accuracy.

In the original version of the manuscript I found the text to be overlong and with redundancies that detracted from the quality of the study. Again, improvements have been made. As it now stands, the manuscript reads well, with a few exceptions detailed below. Importantly, the abstract has been revised so as to provide a more accurate summary of the main findings.

We thank the referee for his/her positive comments on our manuscript.

There are some outstanding issues that must be dealt with:

1. Even though the abstract has been revised and improved, it stills falls short of conveying effectively to the reader the novelty of the present paper. As I see it, the most interesting finding relates to time scales. As referred to in the Introduction, some of the environmentally induced changes in OB function are known to be rapid while the synaptogenesis of adult-born neurons occurs over a longer time scale. This discrepancy in time scales implies that there must be structural modifications at play that are quicker than those that depend on the development of new synapses. The abstract addresses this issue in a rather oblique fashion. Some additional modifications are required.

We have modified the abstract to mention differences in the timescale between environmental changes and synaptogenesis of adult-born neurons.

2. The abstract states that there is an "unusual form of structural remodeling". The term "unusual" strikes me as odd, even more so since the authors (in the discussion) speculates that this type of spine relocation may occur also in other neuronal networks and thus be a phenomenon that is not restricted to the olfactory system.

We remove the word "unusual" from the abstract and elsewhere in the text.

3. The text needs to be checked carefully for misprints and inconsistencies. Misspellings abound. And there are some unorthodox terms and expressions - like "synaptophysin+" on p. 4, "thrilling to propose" on p 19, and "all this suggest" on p 20. Clearly the manuscript must be subjected to a final check before resubmission.

The manuscript was checked carefully and all the misprints and inconsistencies were corrected.

Referee #4:

The authors observe motile spine head filopodia (SHF) in the granule cell dendrites in the olfactory bulb. They show that a larger total number of SHF (imaged over 4 hours) correlate with greater normalized spine relocation (aka displacement) and greater amplitude of the direction vector of displacement (microns) in the olfactory bulb. These SHF bearing spines are more common and more motile on adult born compared to early born granule cells. Spine relocation is seen as a method for rapid structural plasticity. To examine the mechanism by which spine head motility direction can be related to spine relocation direction, the authors show olfactory stimulation enhances motility (linking it to activity). In acute slices they then find patterned but not random mitral cell activity alters SHF and relocation. Spine head filopodia motility is dependent on AMPA and direction can be related to BDNF. Occlusion and BDNF mitral cell KO interact to alter spine density and % of spines with SHFs. A model is used to suggest % changes observed in vivo could alter olfactory encoding.

This paper contains a wealth of information about mechanisms but could be more clear about the basic finding, what it is and is not, and its relative occurrence compared to other forms of structural plasticity. Namely what is the motility of spines without SHF, spine gains and losses (with and without SHF), and changes in spine volume (with and without SHF). When there is a spine with a SHF is this a hotspot generally for the whole dendrite?

We thank the referee for the constructive comments. As specified below, we performed additional analysis requested by the referee.

1. What is (and is not) relocation? Also displacement (a term used for figures)? The manuscript would greatly benefit from a more clear definition of spine relocation and the criteria for scoring (for inclusion and exclusion). Also how can relocation be distinguished from a situation where a spine is lost and a new one grows nearby or on the other side of a dendrite? How does it differ from spine neck motility in general? Fig. 1d lower row. It is not clear to me that this is an example of relocated spine rather than a new spine (the authors claim to not have ever observed a new spine). Here near the initial arrow there is still a structure remaining on the lower side and to my eye there may be gains on the upper side. It is also unclear if this is the same spine on Fig 3d lower panel on the first day and last (with a gap of imaging in time which cannot be accounted for). As a reader I'd like to know what percentage of relocation events fall into this more ambiguous category? The more ambiguous cases might muddy the data for more conservatively defined events. The methods do not adequately address the subjectivity of the confusion between relocation and new growth for these examples. Cases that show translocation across the dendrite or in which new neighboring spines might also be relocated originals should be excluded to eliminate these ambiguities.

The main criteria for inclusion of spines into the analysis were: 1) spine should be easily identified from other spines on the dendritic segment, and 2) spines need to be identified as stubby or mushroom, i.e. the spine head volume has to be considerable. We excluded only spine located in crowded regions. We understand the reviewer comment with regard to **Figs. 1d and 3d** (lower panel). We thought that these examples would be interesting to report, as they may imply that

some of synapse gain and lost can be in fact explained by spine relocation. We, however, agree with the referee that these examples may be ambiguous and therefore removed all these examples from the manuscript and analysis, as the referee suggested. This does not affect the conclusions of the paper, since we observed only 2 examples out of 23 during time-lapse imaging and 1 example out of 16 during chronic imaging. For all other cases, our time-lapse imaging allows us to clearly see gradual relocation of isolated spine from its initial position to a new position, as it is shown in **Figs. 1d,1f, 3c-d, 4h**, etc.

2. What happens in the local and larger neighborhood of spines with SHF? When spines relocate or have SHF is there also higher nearby hotspot activity compared to the rest of the dendrite? There seem to be small spines in many of the images that are transient or part of a compound that become discriminable. This is despite methods suggesting crowded regions were avoided.

We thank the referee for suggesting this analysis. We quantified SHF dynamic on spines located close to ($<15\mu\text{m}$) and far away ($>15\mu\text{m}$) of relocating and non-relocated (devoid of SHF) spines. Our new analysis has indeed revealed that spine relocation occurs on the dendritic segments showing higher local structural plasticity. These data are now shown in the **Supplementary Fig. 2d-f**.

3. What does the motility of spines that do not have SHF look like when compared to spines that do? Do non SHF spines relocate too? This is perhaps captured by Fig 1i but could be discussed more clearly and could be explicitly compared to spines with SHF.

We performed additional analysis on spines devoid of SHF and compared it to spines with SHF and spines with SHF relocating above z-score. These results are shown in the **Fig. 2e** and demonstrate that spines without SHF do not relocate. Only one spine that was devoid of SHF has relocated (**Fig. 2d**). In contrast, 95.3% of spines that relocated above z-score had SHF.

4. What spine volume changes (measured by intensity normalized to backbone) occur with relocation? Do they dim before relocation or do they always grow? Also does this occur to neighbors?

We measured changes in the volume (assessed as changes in the fluorescence) of relocating and non-relocating spines, as the referee suggested and did not observed any differences during the relocation process as well as in comparison to non-relocating spine. These data are shown in the **Supplementary Fig. 1d**.

5. Writing and interpretation of causality is an issue.

a)The abstract states: "This relocation is induced by the growth of spine head filopodia (SHF) and is driven by principal cells activity." The interpretation here could be toned down. Also replace "preexisting" GCs with more specific descriptor like early born.

We modified the abstract to integrate these changes.

b) Discussion of Fig 8 (in results and discussion) should consider alternate orders of

causal events. Especially if SHF are by definition transient to the time of the experiment, they may simply tend to be present later on spines that survive, possible spines close to a source of BDNF, and SHF themselves may not play a causal role in spine survival.

We re-wrote this part to take into account the remarks of the referee.

c) Language in the discussion of the model also suggest experiments were actually done "We also show that spine relocation contributes to odor information processing and allows for fast synchronization of MC in olfactory learning paradigms."

We specified that these are modeling experiments.

6. When showing "n" this is often SHF or spines. Please state which and also indicate in how many individual cells and mice.

We now specified if n is SHF or spines and mentioned the number of individual cells and mice in the manuscript and Figure legends.

7. Panel II does not fully address Reviewer 2 question 4. This should also be done on spines with SHF to address this question: "How much of the SHFs turned into spine head relocation in various experimental conditions examined (Fig. 1-7)."

We now report in the manuscript how much of the SHF turned into spine relocation for different experimental conditions. For 90 min baseline condition 6.1% of SHF turned into spine relocation. This percentage increases from 3.7 % (30 min baseline imaging) to 25.9 % after 30min of stimulation of mitral cells (**Fig. 4n**) and 38.1% following BDNF puff application (45 min) (**Fig. 6g**). The difference in the percentage of relocating spines between BDNF puff and electrical stimulation of mitral cells is explained by longer period of imaging for BDNF experiments that gives more time for spines to relocate above the threshold. No spine relocation was observed for the experiments described in the **Figs. 5, 7**.

8. Fig 3 addresses reviewer #3 request but should also compare to non relocating spines.

We compared to non-relocating spines, as the referee suggested, and observed that 90.7% of stable spines were found on the second imaging day.

9. Reviewer #3 also asked about nearby vs. distal spines and the impact of stimulation (Is the stimulation generally potent to elicit changes near stimulation, or it more global to the connected neuron? Actually, what is the distance of effective activation here?) In the rebuttal the authors point to a figure on SHF lifetime (4i), to suggest the effect on spine relocation is local. Similar data showing spine relocation would make this point.

We performed similar analysis for nearby and distal spines. These data are shown in the **Fig. 4p** and confirm that the effect is local.

10. New figure 3 images show relocating spines but these do not have SHF (or they are

not clear). Were they present in other images? Also same issue for 4h. Also Fig 3, the images showing re-identification of a dendrite in fixed tissue is also not convincing. The spines do not match up. This also begs what data connects relocation exclusively to SHF possession? This is relevant to Reviewer 2 question 4 above in item 7.

SHF were present on the same spines during different time points. We modified the figure to show SHF. We also show low magnification images for **Fig. 3** to demonstrate that imaged dendrites matched the ones used for immunohistochemistry. Several lines of evidence suggest that spine relocation is linked to SHF. First, 95.6% of relocating spines had SHF; second, direction of SHF growth determines direction of spine relocation; and third, the number of SHF correlates with spine relocation. Only one spine that showed relocation was devoid of SHF, but even in this case we cannot exclude the possibility that we missed SHF between imaging sessions because of rapid appearance and disappearance of SHF.

11. Spine relocation (or displacement as it is called here) is likely to grow with time. So it seems a priori that plots like 4o showing a relocation bias in the second period are also likely to show a bias under control conditions. However random stimulation experiments added in this round don't find this. The control experiment plot should perhaps be considered for the main manuscript and the discrete timing of relocation could be discussed.

We included the graph with random stimulation into the main figures (**Fig. 4o**) and discussed that relocation is not due to different timing. We also would like to mention that we did control experiments to ensure that rapid structural modifications observed in our study are stable over time. To show this, we quantified the lifetime and dynamic of SHF during 90 min imaging session and observed the same structural dynamic during the first and second 45 min of imaging (**Supplementary Fig. 4b**).

12. Was the data scored blind to condition?

The data was not scored blind to condition.

Reviewers' Comments:

Reviewer #4 (Remarks to the Author)

The authors have been responsive to the previous review.

However, I am still getting clear on 'What is relocation?' Is this truly a new form of structural plasticity? It should be discussed if it is possibly a new version of description of motility of spine necks? Or spine volume? Can major refs for these be added to discussion. Is it necessarily also a change in orientation? Therefore, it is important to know 'Do relocating spines change their necks or head or neck volume as spines do under LTP paradigms?' Could this be the reason they visually relocate? This issue was part of my previous review. New Fig S1d suggests that the relocating spines start at smaller volumes than non-relocating spines, but the relocation event timing is not indicated here and no effect is indicated. This should be clarified and presented more clearly with aligned events. Also it would also help to know are any relocation events observed (or ignored) involving a shortening of the neck?

The SHF data are far more clear to me and interesting as a precursor to the relocation event once it is better defined.

I particularly appreciate the new information on sampling and frequency of observations.

I am very concerned however that the data were not analyzed blind to condition. A small inadvertent bias in favor of a single spine could have large effects. This can be seen in the new details provided.

Line 224 "Moreover, stimulating the MC with a physiological pattern increased the percentage of spines with SHF that relocated above the threshold from 3.7 % to 25.9 % (30 min imaging under baseline and stim. conditions, n = 27 spines, 11 MC-GC pairs, 9 mice)."

Pg 6-7 Causality and sufficiency arguments on line 122-124, 132 etc. are problematic for me. I think it is too much to claim that SHF were required for relocation. Data so far here suggest they co-occur. They however have not been blocked or induced yet. This argument could be perhaps be better built in the discussion once all the data is presented.

The figures show AMPA stabilizing the SHF. But on line 404 in discussion it says they stabilize spines. Is relocation blocked too? Fig 5h shows and AMPA effect on relocation is not significant. Blocking a rare event is hard to show. I am still unclear on the model, more stable SHFs would be expected to do what to relocation? Enhance or it or reduce it?

Fig S2. Shows there may be hotspots for SHF formation (not relocation) within 15 um. It is not clear if raw distance matters more or if it also matters if the spine share a parent branch. Could this be clarified? Also the description of what is being measured could be written more clearly in the text. SHF formation instead of "structural dynamics."

For S4 it is confusing when the SHF can appear and then be counted. From the S4 figure arrow 2, I wonder is there any advantage to appearing in baseline allowing longer observation? Therefore, should SHF that appear during a stim or drug phase be quantified differently than those that appear beforehand in baseline since we cannot know when they stabilize? Also Looking at S4d how is the concluding legend "Activation of AMPA is required for SHF motility" when the antagonist NBQX enhances motility. From the data I would say Activation of AMPA is required for SHF stability.

Minor

Fig S3b Y axis says minutes and should be a ratio
line 167 use of mediated?

Line 21 need two sentences instead of one?

Line 368 discover should perhaps be describe (ep since you say others described SHF below)

Overall delaying and lightening causality arguments would help. Text might also also weight SHF description as much as spine relocation.

P375 modeling suggests but does not show

Reviewer #4 (Remarks to the Author)

The authors have been responsive to the previous review.

However, I am still getting clear on 'What is relocation?' Is this truly a new form of structural plasticity? It should be discussed if it is possibly a new version of description of motility of spine necks? Or spine volume? Can major refs for these be added to discussion. Is it necessarily also a change in orientation? Therefore, it is important to know 'Do relocating spines change their necks or head or neck volume as spines do under LTP paradigms?' Could this be the reason they visually relocate? This issue was part of my previous review. New Fid S1d suggests that the relocating spines start at smaller volumes than non-relocating spines, but the relocation event timing is not indicated here and no effect is indicated. This should be clarified and presented more clearly with aligned events. Also it would also help to know are any relocation events observed (or ignored) involving a shortening of the neck?

Several features distinguish the spine neck or volume changes observed following long-term potentiation (LTP) from spine relocation. First, the spine neck/volume changes observed following LTP occur in the nanometer range (Tonnesen et al., 2014; Araya et al., 2014), whereas the spine relocations we observed occurred in the micrometer range and were as large as 2-4 μm (**Fig. 1c-f**). Second, spine neck plasticity has been described as a shortening and widening of the spine neck (Tonnesen et al., 2014; Araya et al., 2014), whereas in the present study most spine relocation resulted from a lengthening of the spine neck (16 of 21 spines). Third, spine relocation was often associated with changes in orientation (12 spines out of 21), which is not observed during spine neck plasticity induced by LTP (Tonnesen et al., 2014; Araya et al., 2014). In addition to the cases observed under baseline conditions, this can also be seen in **Figs. 4-6**, which show clear reference points (dendrite of a stimulated mitral cell, or puff pipette) toward which SHF and spines relocate (i.e., changes in orientation). We have added a description of these differences to the Results section.

We also observed no differences in spine volumes during relocation process (**Supp. Fig. 1e**). As suggested by the reviewer, we aligned relocating spines with their maximum displacement event (time-point 0 min in **Supp. Fig 1f**) and, once again, observed no significant differences in spine volume (**Supp. Fig. 1g**).

These results indicated that spine relocation and spine neck/volume plasticity are different forms of structural modifications.

The SHF data are far more clear to me and interesting as a precursor to the relocation event once it is better defined.

I particularly appreciate the new information on sampling and frequency of observations.

I am very concerned however that the data were not analyzed blind to condition. A small inadvertent bias in favor of a single spine could have large effects. This can be seen in the new details provided.

Line 224 "Moreover, stimulating the MC with a physiological pattern increased the percentage of spines with SHF that relocated above the threshold from 3.7 % to 25.9 % (30 min imaging under baseline and stim. conditions, n = 27 spines, 11 MC-GC pairs, 9 mice)."

The main findings of our study on spine relocation were determined under baseline conditions (**Figs. 1-3, supp. Figs. 1-3; Suppl. Movies 1 and 2**) and, as such, there were no experimental conditions. To perform an unbiased analysis of the data, we randomly tracked a large population of spines (514 spines) and used statistical methods such as z-scores of direction vector amplitudes (**Fig. 1i**), allowing for stringent, statistical definition of spine relocation. Spine relocation was thus not defined by the experimenter.

For the experiments revealing the molecular mechanisms of spine relocation, we would like to emphasize that the same spines were tracked under baseline and experimental conditions and that, once again, statistical values were used to define spine relocation. This approach was based on quantitative automatic measurements of spine movements using a preset threshold value and not a qualitative assessment of the number of spines that relocate. The preset threshold value was consistent across all conditions. We were confused by the reviewer's suggestion that the observed effect might have been due to changes to a single spine and by the example provided to support this suggestion. In this particular example, the 7-fold change could not have been due to a single spine. Furthermore, no increase was observed in the control experiments using a random stimulation pattern.

For other quantifications involving SHF dynamics and lifetime, the data were averaged per cell to avoid any "spine" bias.

Altogether, this indicated that these data were not biased by the experimenter being aware of the experimental condition.

Pg 6-7 Causality and sufficiency arguments on line 122-124, 132 etc. are problematic for me. I think it is too much to claim that SHF were required for relocation. Data so far here suggest they co-occur. They however have not been blocked or induced yet. This argument could be perhaps be better built in the discussion once all the data is presented.

We modified this text and removed the causality and sufficiency arguments from the Results section.

The figures show AMPA stabilizing the SHF. But on line 404 in discussion it says they stabilize spines. Is relocation blocked too? Fig 5h shows and AMPA effect on relocation is not significant. Blocking a rare event is hard to show. I am still unclear on the model, more stable SHFs would be expected to do what to relocation? Enhance or it or reduce it?

We propose that the active sensing of SHF, as reflected by their rapid protrusion/retraction, is an integral part of spine head relocation. **Fig. 2d** shows that the average number of SHF is 2.95 ± 0.4 for relocating spines and 1.4 ± 0.07 for non-relocating spines. We propose that under baseline conditions SHF “explore” the microenvironment for factors (glutamate, BDNF) released in an activity-dependent manner. The concurrent action of these factors leads to SHF stabilization (AMPA) and spine relocation (BDNF). Thus, the decreased number of SHF (**Fig S4e**), as reflected by SHF stabilization (**Fig 5f**), impede spine relocation following the application of AMPA (**Fig 5h**). Therefore, we did not quantify a “blocking of rare event,” but rather the absence of any increase in spine relocation following the application of AMPA. We used a similar experimental design to apply BDNF and spine relocation increased from 0% under baseline conditions to 38.1% following the BDNF application. If AMPA induced a similar or an even smaller increase this would have been detected by our experiments and analyses.

Fig S2. Shows there may be hotspots for SHF formation (not relocation) within 15 um. It is not clear if raw distance matters more or if it also matters if the spine share a parent branch. Could this be clarified? Also the description of what is being measured could be written more clearly in the text. SHF formation instead of "structural dynamics."

We split our data into two groups, that is, SHF on spines sharing same the parent dendrite and SHF on spines belonging to different dendrites of the same cell. While we did not observe any differences between the two groups (see figure attached), the number of observation is low. In many of our experiments we imaged spine on rather isolated dendrites to reliably track spine relocation. Because of that, we do not have many cases of relocating spine for which parent and non-parent dendritic branches were simultaneously imaged. This precluded us for including these data into the manuscript. We also modified the figure legend to address the reviewer’s concerns.

For S4 it is confusing when the SHF can appear and then be counted. From the S4 figure arrow 2, I wonder is there any advantage to appearing in baseline allowing longer observation? Therefore, should SHF that appear during a stim or drug phase be quantified differently than those that appear beforehand in baseline since we cannot know when they stabilize? Also Looking at S4d how is the concluding legend "Activation of AMPA is required for SHF motility" when the antagonist NBQX enhances motility. From the data I would say Activation of AMPA is required for SHF stability.

SHF that appeared during baseline conditions and that were maintained during the application of the drug were not quantified. We only took the SHF that appeared during one block of the experiment (either BL, STIM, Drug+STIM) into consideration. This has been specified in the Method section. We also changed the figure legend and title to address the reviewer's concerns.

*Minor
Fig S3b Y axis says minutes and should be a ratio*

Yes, thank you. It is a ratio and not minutes.

line 167 use of mediated?

We changed the word "mediated".

Line 21 need two sentences instead of one?

Line 21 is the title page.

Line 368 discover should perhaps be describe (ep since you say others described SHF below)

We changed "discover" to "reveal". To the best of our knowledge spine relocation has never been previously reported.

Overall delaying and lightening causality arguments would help. Text might also also weight SHF description as much as spine relocation.

We revised the manuscript based on the reviewer's suggestions.

P375 modeling suggests but does not show

We changed "show" to "suggest".